# RNase III-mediated processing of a *trans*-acting bacterial sRNA and its *cis*-encoded antagonist

Sarah Lauren Svensson, Cynthia Mira Sharma*

Department of Molecular Infection Biology II, Institute of Molecular Infection Biology, University of Würzburg, Würzburg, Germany

**ABSTRACT** Bacterial small RNAs (sRNAs) are important post-transcriptional regulators in stress responses and virulence. They can be derived from an expanding list of genomic contexts, such as processing from parental transcripts by RNase E. The role of RNase III in sRNA biogenesis is less well understood despite its well-known roles in rRNA processing, RNA decay, and cleavage of sRNA-mRNA duplexes. Here, we show that RNase III processes a pair of *cis*-encoded sRNAs (CJnc190 and CJnc180) of the food-borne pathogen *Campylobacter jejuni*. While CJnc180 processing by RNase III requires CJnc190, RNase III processes CJnc190 independent of CJnc180 via cleavage of an intramolecular duplex. We also show that CJnc190 directly represses translation of the colonization factor PtmG by targeting a G-rich ribosome-binding site, and uncover that CJnc180 is a *cis*-acting antagonist of CJnc190, indirectly affecting *ptmG* regulation. Our study highlights a role for RNase III in sRNA biogenesis and adds *cis*-encoded RNAs to the expanding diversity of transcripts that can antagonize bacterial sRNAs.

## Editor's evaluation

*Campylobacter jejuni* is a serious food-borne pathogen and understanding how the various products necessary for its pathogenesis are regulated is a key step in preventing its growth and/or treating disease. Here, Svensson and Sharma examine the complex pathway that leads to the maturation of two complementary regulatory RNAs and how one of the RNAs antagonizes the other to relieve repression of a virulence-related gene.

*For correspondence:
cynthia.sharma@uni-wuerzburg.de

Competing interest: The authors declare that no competing interests exist.

## Introduction

Bacterial small, regulatory RNAs (sRNAs) are an important class of post-transcriptional gene expression regulators that control adaptation to changing environmental conditions or stresses (*Storz et al., 2011*), or can also regulate virulence genes in pathogens (*Quereda and Cossart, 2017*; *Svensson and Sharma, 2016*; *Westermann, 2018*). They are also intimately associated with RNA-binding proteins (RBPs), such as RNA chaperones, as well as ribonucleases that are required for their maturation, stability, degradation, and/or function (*Holmqvist and Vogel, 2018*; *Quendera et al., 2020*). While most of the first identified and characterized sRNAs in bacterial genomes are expressed as stand-alone, intergenically encoded transcripts, genome-wide RNA-seq based approaches have identified sRNAs hidden in unexpected genomic contexts, including in 5'/3' untranslated regions (UTRs) and in coding regions of mRNAs, or even in housekeeping RNAs (reviewed in *Adams and Storz, 2020*). These include members of the expanding class of 3' UTR-derived sRNAs mainly studied in Gammaproteobacteria (*Miyakoshi et al., 2015a*). These can be transcribed from an independent promoter, such as *Escherichia coli* MicL (*Chao et al., 2012*; *Guo et al., 2014*), or can be processed

from mRNAs by the single-stranded RNA endonuclease RNase E (*Chao and Vogel, 2016*; *De Mets et al., 2019*). Intergenically encoded, stand-alone sRNAs can also require maturation by RNase E to increase their stability (*Chae et al., 2011*; *Davis and Waldor, 2007*; *Hör et al., 2020*), to expose their seed regions (*Papenfort et al., 2009*), or even to create two different sRNAs with distinct regulons (*Fröhlich et al., 2016*; *Papenfort et al., 2015*).

Despite progress in defining mechanisms and regulatory consequences of complex sRNA biogenesis pathways in model Gammaproteobacteria, less is known about how such sRNAs are generated in bacteria lacking RNase E. RNase E has the most well-characterized role in bacterial sRNA biogenesis (*Bandyra and Luisi, 2018*; *Miyakoshi et al., 2015a*) but is absent in ~1/5 sequenced strains (*Hui et al., 2014*). In contrast, bacteria almost universally encode RNase III (*Court et al., 2013*). RNase III recognizes double-stranded RNA (11–20 base-pairs long) in a mostly sequence-independent manner, and cleaves both strands to generate characteristic 2–3 nucleotide (nt) 3′-overhangs. Single-strand nicking, especially at imperfect duplexes, can also occur (*Altuvia et al., 2018*; *Court et al., 2013*; *Le Rhun et al., 2017*). Bacterial RNase III is mainly known for its role in rRNA processing, in maturation or decay of certain mRNAs, and in cleavage of sRNA-mRNA duplexes (*Court et al., 2013*). While the RNase III domain-containing ribonucleases Dicer and Drosha play a central role in sRNA biogenesis in eukaryotes (*Carthew and Sontheimer, 2009*), the role of bacterial RNase III in sRNA biogenesis is less clear. In *Staphylococcus aureus*, RNase III generates RsaC sRNA from the 3′ UTR of *mntABC* mRNA (*Lalaouna et al., 2019*). Its expropriation for biogenesis of CRISPR RNAs (*Deltcheva et al., 2011*; *Dugar et al., 2018*), as well as genome-wide studies of the RNase III targetome that report sRNAs as potential targets in Gram-negative and Gram-positive species (*Altuvia et al., 2018*; *Gordon et al., 2017*; *Le Rhun et al., 2017*; *Lioliou et al., 2013*; *Lioliou et al., 2012*; *Lybecker et al., 2014*; *Rath et al., 2017*), indicate that RNase III might process sRNAs in diverse bacteria. However, most of these potential RNase III targets remain to be validated or studied.

In addition to an expanding genomic context of regulatory RNA sources, there is also emerging evidence for high complexity of bacterial post-transcriptional networks involving not only cross-talk with transcriptional control, but also RNA antagonists that can sequester and modulate RBPs (*Dugar et al., 2016*; *Romeo and Babitzke, 2018*; *Sterzenbach et al., 2013*; *Wassarman, 2018*) or even regulate stability or function of other RNAs as so-called competing endogenous RNAs (ceRNAs), RNA decoys/predators, or sponge RNAs (*Figueroa-Bossi and Bossi, 2018*; *Grüll and Massé, 2019*; *Kavita et al., 2018*). Such RNA antagonists can be derived from diverse cellular transcripts, including mRNAs (UTRs and coding regions) (*Adams and Storz, 2020*; *Adams et al., 2021*; *Figueroa-Bossi et al., 2009*; *Miyakoshi et al., 2015b*) and tRNAs (*Lalaouna et al., 2015*), or can be stand-alone sRNAs encoded in the core genome or in prophages (*Bronesky et al., 2019*; *Melamed et al., 2020*; *Tree et al., 2014*). Unbiased global biochemical and genetic screens for sRNA expression and regulation have recently recovered several characterized examples of *trans*-acting sRNA antagonists in Gram-positive and Gram-negative bacteria (*Bronesky et al., 2019*; *Chen et al., 2021*; *Durand et al., 2021*; *Melamed et al., 2020*), including those affecting infection phenotypes via antagonism of central sRNA regulators of virulence (*Le Huyen et al., 2021*). Despite reports of extensive antisense transcription in diverse bacteria and the demonstrated role of *cis*-encoded antisense RNAs (asRNAs) in control of mRNA translation and stability (*Thomason and Storz, 2010*), less is known about whether RNAs encoded in *cis* to other sRNAs can act also as antagonists and how they might affect the biogenesis, stability, or function of their antisense sRNA partners.

The zoonotic pathogen *Campylobacter jejuni* is currently the leading cause of bacterial food-borne gastroenteritis worldwide (*Burnham and Hendrixson, 2018*; *Havelaar et al., 2015*). How *C. jejuni* regulates its gene expression to adapt to different environments is so far unclear, as its genome encodes only three sigma factors and lacks homologs of certain global stress response regulators such as RpoS (*Parkhill et al., 2000*; *Young et al., 2007*) as well as of the global RNA chaperones Hfq and ProQ (*Pernitzsch and Sharma, 2012*; *Quendera et al., 2020*). Our comparative differential RNA-seq (dRNA-seq) analysis of multiple *C. jejuni* strains revealed many conserved and strain-specific sRNAs and asRNAs (*Dugar et al., 2013*). However, functions are still largely unknown for most of these. Besides so far not encoding a general sRNA chaperone, it is also unclear which RNases participate in sRNA biogenesis and function in *C. jejuni*. Although Epsilonproteobacteria such as *C. jejuni* and the related gastric pathogen *Helicobacter pylori* are Gram-negative, they surprisingly encode an RNase repertoire more similar to Gram-positives: for example, RNase Y and RNase J instead of RNase E

(*Parkhill et al., 2000*; *Pernitzsch and Sharma, 2012*; *Tomb et al., 1997*). *C. jejuni* also encodes an RNase III homolog (*Haddad et al., 2013*), which participates in the biogenesis of Type II-C CRISPR RNAs (*Dugar et al., 2018*; *Dugar et al., 2013*). Beyond this and a role in rRNA biogenesis, the function of *C. jejuni* RNase III is unclear.

Here, we have characterized the biogenesis and mode of action of a conserved, processed pair of *C. jejuni cis*-encoded sRNAs, CJnc180/190. We previously reported that deletion of these sRNAs affects *C. jejuni* virulence in a three-dimensional tissue-engineered model of the human intestine (*Alzheimer et al., 2020*). While this seemed to be mediated at least in part via repression of the flagellin modification factor PtmG, the molecular mechanisms underlying this, the roles of each of the two sRNAs in PtmG regulation, as well as how they are processed remained unknown. Here, we demonstrate that both RNAs are processed by RNase III and that mature CJnc190 directly represses translation of *ptmG* mRNA by base-pairing with a G-rich sequence over its RBS. Surprisingly, although both RNAs are encoded antisense to each other and show perfect complementarity, suggesting they might be co-processed by RNase III, only CJnc180 requires its antisense partner for maturation. Instead, CJnc190 is transcribed as longer precursors which can fold into extended stem-loop structures that are processed by RNase III independently of CJnc180. Finally, we demonstrate that CJnc180 is a *cis*-acting antagonist of CJnc190. Overall, our characterization of CJnc180/190 demonstrates a role for RNase III in sRNA maturation and also reveals the potential for *cis*-encoded sRNA-sRNA targeting.

## Results

### A product of the CJnc180/190 sRNA locus represses expression of *ptmG*

Our comparative dRNA-seq study of multiple *C. jejuni* isolates revealed a conserved pair of antisense sRNAs, CJnc180 and CJnc190 (annotated as 99 and 216 nt, respectively, in strain NCTC11168) (*Dugar et al., 2013*; *Figure 1A*). RNA-seq patterns and northern blot analysis suggested that both sRNAs might be processed from longer forms (*Dugar et al., 2013*). While CJnc180 was detected in wild-type (WT) as both an ~90 nt 'mature' 5′ end-derived species and an ~160 nt longer putative precursor (pre-CJnc180) on northern blots, only a single CJnc190 RNA (~70 nt) was detected. This CJnc190 species appeared to arise from the 3′ end of its annotated primary transcript (pre-CJnc190). Mature CJnc180 (hereafter, CJnc180) shows almost complete complementarity to mature CJnc190 (hereafter, CJnc190, *Figure 1A*).

We previously observed that deletion of CJnc180/190 affects *C. jejuni* adherence and internalization in our Caco-2 cell-based tissue-engineered model of the human intestine (*Alzheimer et al., 2020*), suggesting that CJnc180 and/or CJnc190 regulate genes involved in *C. jejuni* virulence. Analysis of total protein profiles by SDS-PAGE revealed a ~45 kDa band upregulated upon deletion of CJnc180/190 (Δ180/190) (*Figure 1B*), which was identified as the colonization/infection-relevant flagellin modification factor PtmG (Cj1324) (*Alzheimer et al., 2020*; *Howard et al., 2009*) based on mass spectrometry (*Figure 1—figure supplement 1*; *Supplementary file 1a*). The upregulated band was no longer observed in a Δ180/190 Δ*ptmG* double mutant, confirming it as PtmG (*Figure 1B*). Complementation of Δ180/190 with a region spanning the Cj1650 and *map* intergenic region (C-180/190, dashed line, *Figure 1A*) at the unrelated *rdxA* locus rescued expression of both sRNAs and restored PtmG repression (*Figure 1B*).

Northern blot analysis further demonstrated that not only the protein level but also *ptmG* mRNA levels are upregulated almost 10-fold in Δ180/190 vs. WT and are reduced (2-fold) upon overexpression of CJnc180/190. Overall, these data indicate that at least one of the two sRNAs is involved in repression of the gene encoding the PtmG colonization factor.

### The mature CJnc190 sRNA is sufficient to repress *ptmG*

To start to disentangle the roles of each sRNA in potential direct regulation of *ptmG*, we first defined the 5′ and 3′ ends of each mature sRNA in WT using primer extension and 3′ RACE (rapid amplification of cDNA ends), respectively (for details, see *Figure 1—figure supplement 2* and *Figure 1—figure supplement 3*). Subsequent predictions of secondary structures and potential *ptmG* mRNA interactions with the mature sRNA sequences revealed a strong potential for a C/U-rich loop within CJnc190 to base-pair with the ribosome-binding site (RBS) and start codon of *ptmG* mRNA (*Figure 1C*). A less

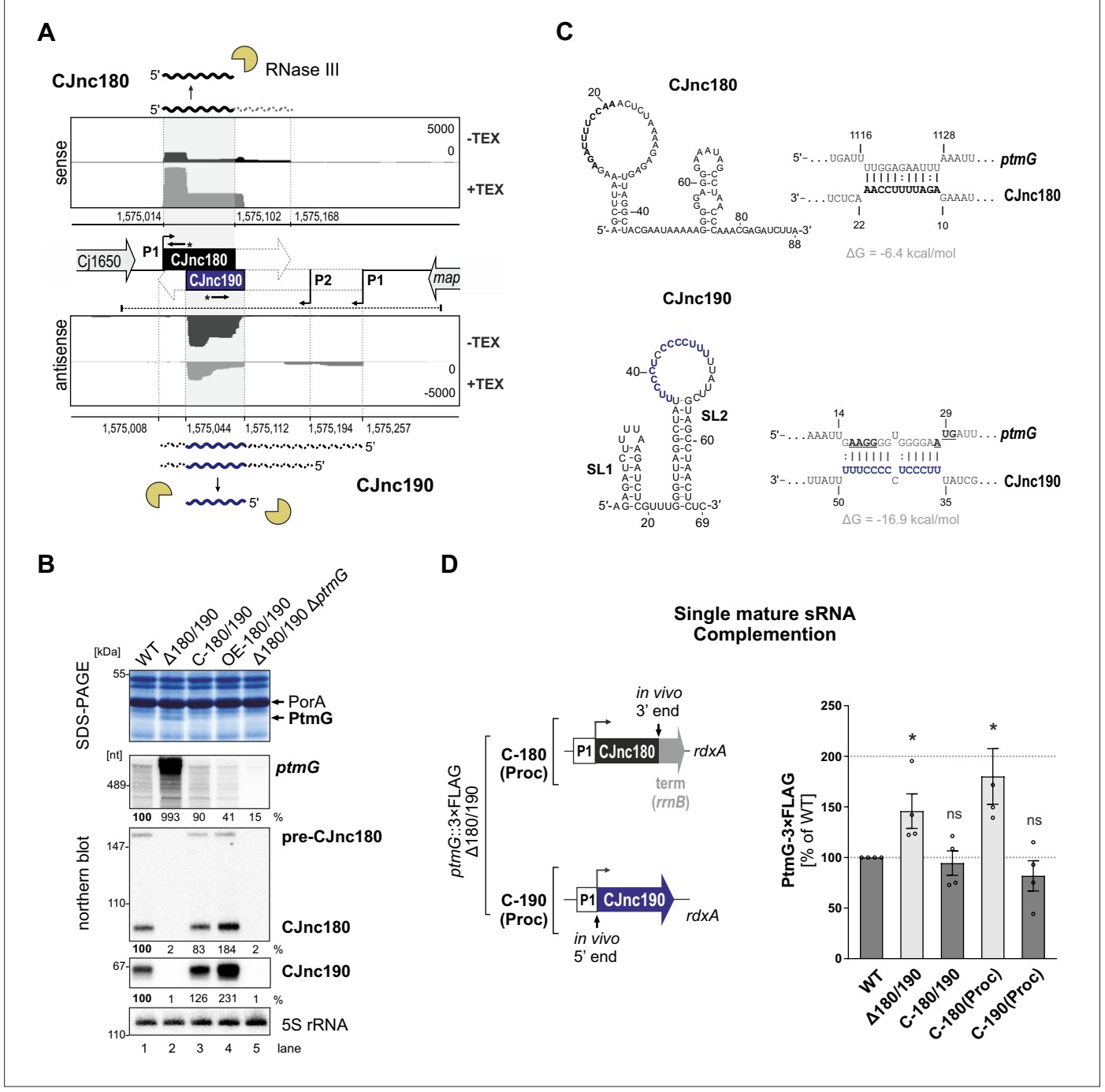

**Figure 1.** The processed CJnc190 sRNA represses *ptmG* expression. (**A**) Differential RNA-seq (dRNA-seq) coverage for CJnc180 and CJnc190 sRNAs in *Campylobacter jejuni* NCTC11168 (***Dugar et al., 2013***). -/+ TEX: mock/treated terminator exonuclease (TEX) dRNA-seq libraries. TEX treatment degrades processed (non-triphosphorylated) 5′ ends, enriching 5′-triphosphorylated primary transcript ends at transcription start sites (TSS). Bent arrows: TSS. Black dashed line: genomic region used for complementation (C-180/190). Starred arrows: northern blot probes for mature sRNAs. P1/P2: promoter motifs. (**B**) SDS-PAGE and northern blot analyses of total protein and RNA, respectively, from *C. jejuni* wild-type (WT) and sRNA/*ptmG* mutant strains. Upregulated ~45 kDa PtmG and non-regulated PorA control are indicated. Probes for the mature sRNAs (CSO-0189/0185 for CJnc180/190), respectively (starred arrows, panel A), and the 5′ end of the *ptmG* ORF (CSO-1666) were used. As a loading control, 5S rRNA was probed with CSO-0192. OE, second-copy overexpression. (**C**) Predicted secondary structures (RNAfold) (***Lorenz et al., 2011***) and *ptmG* interactions (IntaRNA) (***Mann et al., 2017***) for mature CJnc180 and CJnc190. Bold/blue: potential *ptmG* pairing residues for CJnc180/CJnc190, respectively. Underlined: *ptmG* RBS/ start codon. (**D**) Complementation of *ptmG* regulation in Δ180/190 with single mature sRNAs. (*Left*) To express CJnc180 only (C-180(Proc)), its mature 3′ end was fused to the *Escherichia coli rrnB* terminator; transcription is driven from its native promoter. For C-190(Proc), the mature CJnc190 5′ end was

*Figure 1 continued on next page*

*Figure 1 continued*

fused to its annotated TSS (P1 promoter) and 125 nucleotides upstream. (*Right*) PtmG-3×FLAG levels measured by western blotting. Error bars represent standard error of the mean (SEM) of four independent replicates. *: p < 0.05, ns: not significant, vs. WT. See also *Figure 1—figure supplement 4A*.

The online version of this article includes the following source data and figure supplement(s) for figure 1:

**Source data 1.** Full northern and western blot images for the corresponding detail sections shown in *Figure 1*, raw data for western blot quantifications, and mature CJnc180/CJnc190 and *ptmG* target sequences used for interaction predictions.

**Figure supplement 1.** Identification of CJnc180/190 targets by SDS-PAGE and mass spectrometry analysis.

**Figure supplement 1—source data 1.** Full image of SDS-PAGE used for mass spectrometry analysis.

**Figure supplement 2.** Mapping 5′ boundaries of mature CJnc180 and CJnc190 by primer extension.

**Figure supplement 2—source data 1.** Full northern blot and primer extension images for the corresponding detail sections shown in *Figure 1—figure supplement 2*.

**Figure supplement 3.** Identification of CJnc180/CJnc190 3′ ends by RACE (rapid amplification of cDNA ends).

**Figure supplement 4.** Post-transcriptional regulation of *ptmG* by CJnc190.

**Figure supplement 4—source data 1.** Full northern and western blot images for the corresponding detail sections shown in *Figure 1—figure supplement 4*.

**Figure supplement 4—source data 2.** Full northern and western blot images for the corresponding detail sections shown in *Figure 1—figure supplement 4*.

stable interaction was predicted between CJnc180 and the 3′ end of the *ptmG* coding region ($\Delta G$ = –6.4 kcal/mol compared to –16.9 kcal/mol for CJnc190:*ptmG*), suggesting that CJnc190, and not CJnc180, directly represses *ptmG* translation. To test this experimentally, we constructed Δ180/190 complementation strains expressing either mature CJnc180 or CJnc190 alone (hereafter C-180(Proc) or C-190(Proc)) from their annotated native promoters (P1, *Figure 1D*, *left*) and measured rescue of *ptmG* repression. A chromosomally epitope-tagged PtmG-3×FLAG fusion was upregulated 1.5-fold upon deletion of CJnc180/190 and rescued to WT levels in the C-180/190 complemented strain (*Figure 1D*, *right*). In line with the prediction that *ptmG* translation is repressed by base-pairing with CJnc190 and not CJnc180, the C-190(Proc) complementation strain had PtmG-3×FLAG levels comparable to WT, whereas C-180(Proc) did not restore regulation of PtmG-3×FLAG. A similar trend was seen for *ptmG* mRNA levels (*Figure 1—figure supplement 4A*). Moreover, a translational GFP reporter fusion of the 5′ UTR and first 10 codons of *ptmG* (*ptmG(10th)*-GFP) was repressed by CJnc190 when transcribed from either the native *ptmG* promoter or the unrelated σ[28] (FliA)-dependent *flaA* promoter, confirming regulation at the post-transcriptional level (*Figure 1—figure supplement 4B*). Collectively, these observations showed that CJnc180 is dispensable and mature CJnc190 is sufficient for post-transcriptional repression of *ptmG*.

## CJnc190 represses *ptmG* translation by base-pairing with its G-rich RBS

We next validated the predicted direct interaction between CJnc190 and *ptmG* mRNA and its requirement for regulation using in vitro and in vivo approaches. These experiments included compensatory base-pair exchanges within the predicted CJnc190:*ptmG* duplex (*Figure 2A*). Gel mobility shift assays using in vitro-transcribed RNAs showed that processed CJnc190 binds the *ptmG* leader, and that a single C-to-G change in its C/U-rich loop (M1) is sufficient to almost completely abolish complex formation (*Figure 2B*). Similarly, a single point mutation (M1′) in the *ptmG* 5′UTR also disrupted interaction with CJnc190, and introduction of the compensatory base exchange in CJnc190 (M1) restored binding. Mature CJnc180 did not bind *ptmG*, in line with the weak predicted CJnc180:*ptmG* interaction. Reciprocal experiments with labeled CJnc190 (WT/M1) and unlabeled *ptmG* leader (WT/M1′) further confirmed the interaction (*Figure 2—figure supplement 1A*). Adding increasing amounts of unlabeled *ptmG* leader (WT) to labeled mature CJnc190 in Inline probing assays protected nucleotides in the C/U rich loop region of 5′ end labeled mature CJnc190 from cleavage, in agreement with the predicted interaction with the *ptmG* leader (*Figure 2A,C*). The same molar ratio of *ptmG* M1′ mutant leader showed less protection, indicating destabilization of the interaction. Reciprocal experiments with labeled WT/M1′ *ptmG* leader and unlabeled WT/M1 CJnc190 confirmed the predicted interaction site on *ptmG* (*Figure 2—figure supplement 1B*).

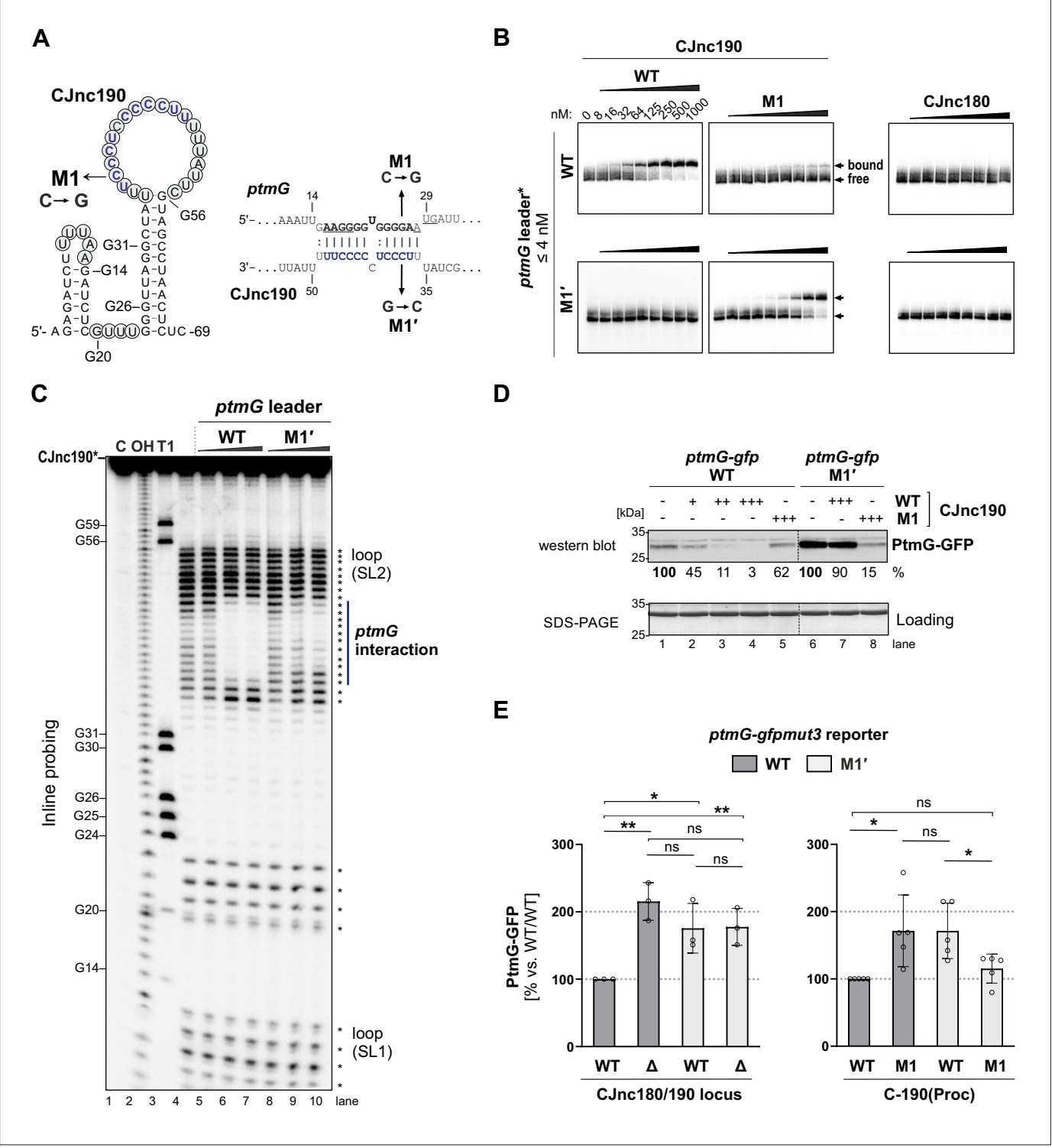

**Figure 2.** CJnc190 represses translation of *ptmG* via base-pairing with its G-rich ribosome-binding site (RBS). (**A**) CJnc190 secondary structure based on Inline probing (panel C) and interaction with the *ptmG* leader showing mutations (M1/M1′) introduced into the interaction site. Circled residues: single-stranded regions mapped by Inline probing. Blue/bold residues: *ptmG*/CJnc190 nucleotides protected in Inline probing (panel A and *Figure 2—figure supplement 1B*). RBS/start codon are underlined. (**B**) In vitro gel shift assay of $^{32}$P-5′-labeled (marked with *) *ptmG* leader (WT/M1′) with unlabeled CJnc190 WT/M1, as well as CJnc180, sRNAs. (**C**) Inline probing of 0.2 pmol $^{32}$P 5′-end-labeled CJnc190 sRNA in the absence or presence of 0.2/2/20 pmol unlabeled *ptmG* leader (WT/M1′). (C) Untreated control; T1 ladder – G residues (indicated on left); OH – all positions (alkaline hydrolysis). (**D**) In vitro translation of a *ptmG(10th)*-GFP reporter (5′ UTR and first 10 codons) of *ptmG* fused to *gfpmut3* (2 pmol) in an *Escherichia coli* cell-free

*Figure 2 continued on next page*

*Figure 2 continued*

system ± CJnc190 (WT/M1; +: 2 pmol, ++: 20 pmol, +++: 100 pmol) detected by western blotting with an anti-GFP antibody. A Coomassie-stained gel of the same samples served as a loading control. (**E**) PtmG(10th)-GFP (WT/M1′) reporter expression in vivo ± mature CJnc190 (WT/M1) measured by western blot analysis. PtmG(10th)-GFP levels are the mean of three (*left*) or five (*right*) independent replicates, with error bars representing the SD. \*\*: p < 0.01, \*: p < 0.05, ns: not significant, vs. wild type (WT). See also *Figure 2—figure supplement 1C*.

The online version of this article includes the following figure supplement(s) for figure 2:

**Source data 1.** Full electrophoretic mobility shift assay (EMSA), Inline probing, western blot, and SDS-PAGE images for the corresponding detail sections shown in *Figure 2*, as well as raw values for western blot quantifications.

**Figure supplement 1.** Direct repression of *ptmG* by CJnc190 via base-pairing.

**Figure supplement 1—source data 1.** Full electrophoretic mobility shift assay (EMSA), Inline probing, western blot, and SDS-PAGE images for the corresponding detail sections shown in *Figure 2—figure supplement 1*, as well as raw values for western blot quantifications.

The observed base-pairing with a G-rich sequence at the *ptmG* RBS suggested that CJnc190 acts by repressing *ptmG* translation. Indeed, the addition of increasing molar ratios (1-, 10-, 50-fold; + to +++ ) of mature CJnc190 to a *ptmG(10th)-gfp* translational reporter mRNA in an in vitro translation system repressed GFP levels in a dose-dependent manner (*Figure 2D*). Consistent with a disrupted interaction, the M1 mutation in the CJnc190 loop partially restored translation of the reporter. While the *ptmG* M1′ mutation alone increased its translation compared to WT, independent of CJnc190 addition, Inline probing experiments did not reveal marked differences in secondary structure for the native (non-GFP-fusion) WT and M1′ *ptmG* leaders in vitro (*Figure 2—figure supplement 1B*). Nonetheless, while the addition of CJnc190 WT did not strongly affect translation of the mutant reporter, addition of CJnc190 M1, carrying the compensatory exchange in its C/U-rich loop, strongly reduced GFP levels generated from the M1′ reporter, indicating restored regulation (*Figure 2D*).

In line with the in vitro results, introduction of the M1′ mutation in vivo in the *ptmG(10th)*-GFP reporter fusion derepressed GFP levels, although not to those of an isogenic Δ180/190 strain (*Figure 2E*, *left*). To account for different levels of CJnc190 in the WT and C-190(Proc) strains, we next compared regulation of the reporter by WT/M1 CJnc190 in the C-190(Proc) background. In line with a disrupted interaction, CJnc190 M1 did not repress the WT *ptmG* reporter to the same levels as CJnc190 WT (*Figure 2E*, *right*), even though WT/M1 sRNAs were similarly expressed (*Figure 2—figure supplement 1C*). Likewise, when WT CJnc190 was expressed with the *ptmG* M1′ reporter, GFP levels were also higher compared to the WT sRNA/leader strain. Finally, in the strain with the compensatory mutations combined (M1/M1′), GFP levels were similar to the isogenic WT/WT (*Figure 2E*). Together, our in vitro and in vivo experiments demonstrate that CJnc190 represses *ptmG* translation via base-pairing with its RBS.

## RNase III processes the CJnc180/190 sRNAs

Next, we set out to gain insight into the biogenesis of the CJnc180/190 sRNA pair. While deletions of most non-essential RNases/RNA degradation enzymes had no major impact on processing of the two RNAs (*Figure 3—figure supplement 1A*), deletion of RNase III (Δ*rnc*) had a dramatic effect on both CJnc180 and CJnc190, abolishing accumulation of the mature sRNA species (*Figure 3A*). For CJnc180, the longer transcript of WT (pre-CJnc180, ~160 nt) was still detected in Δ*rnc*, but at higher levels. For CJnc190, the mature form of WT was completely absent in Δ*rnc* and instead five longer pre-CJnc190 species (~150–280 nt) were detected. All CJnc180 and CJnc190 unprocessed species were absent in a Δ180/190 Δ*rnc* double mutant, ruling out cross-hybridization.

We next asked if RNase III processing of CJnc190 affects *ptmG* regulation. Deletion of *rnc* increased PtmG-3×FLAG protein and mRNA levels to those similar to Δ180/190 (*Figure 3B*, *Figure 3—figure supplement 1B*), and CJnc180/190 deletion in Δ*rnc* did not increase levels further. In contrast to 'native' mature CJnc190 (i.e., processed from precursors), the 'mature' sRNA of the C-190(Proc) strain (transcribed directly from its mature 5′ end) was not markedly affected by *rnc* deletion (*Figure 1—figure supplement 2B*) and was still capable of *ptmG* regulation in the *rnc* deletion background (*Figure 3B*). The mild increase in *ptmG* mRNA levels upon *rnc* deletion in C-190(Proc) suggests that RNase III could also play a minor role in cleavage of CJnc190:*ptmG* duplexes (*Figure 3—figure*

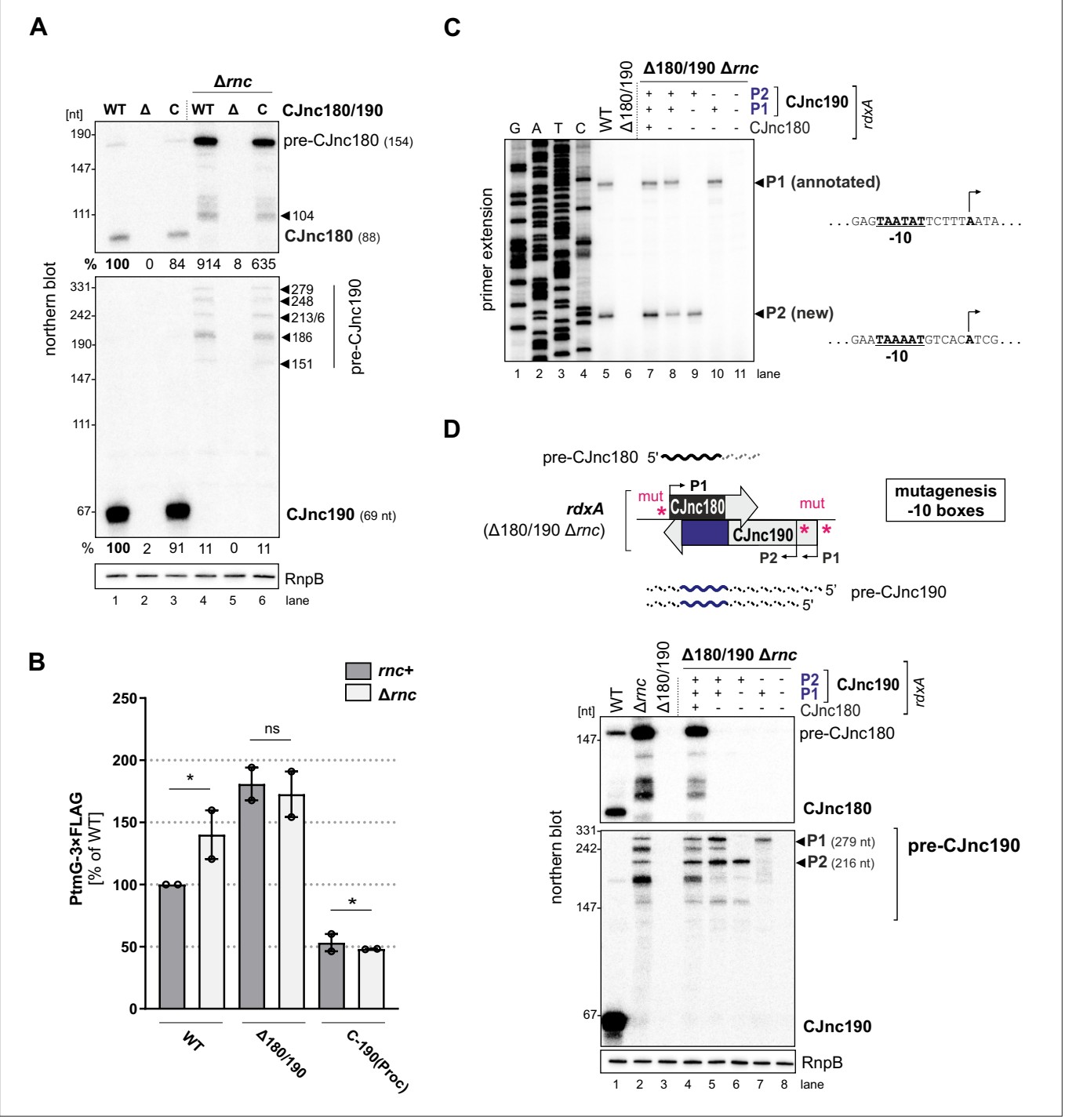

**Figure 3.** RNase III processes CJnc190 precursors expressed from two promoters. (**A**) Northern blot of CJnc180 and CJnc190 processing by RNase III in total RNA. Lengths are based on primer extension and 3' RACE (rapid amplification of cDNA ends). Quantification is for all CJnc190 bands detected in a single strain combined. (**B**) Effect of *rnc* (RNase III) deletion on PtmG-3×FLAG levels in the absence or presence of CJnc180/190 sRNAs. Error bars represent the standard error of the mean (SEM) of two independent replicates. *: p < 0.1, ns: not significant, Student's unpaired t-test. See also **Figure 3—figure supplement 1B**. (**C**) Primer extension analysis of pre-CJnc190 5' ends in wild type (WT) and promoter mutant strains (Δ*rnc* background). Total RNA was annealed with the same probe for mature CJnc190 used for northern blots (CSO-0185). A sequencing ladder was generated with the same probe (lanes 1–4). P1/P2: putative CJnc190 primary transcripts/5' ends. The full gel is shown in **Figure 3—figure supplement 3**. (**D**) Validation of CJnc180/190 promoters. (*Top*) Strategy for testing CJnc180/190 promoter activity by complementation of Δ180/190 with –10 box mutant alleles at *rdxA*. 'mut' – several point mutations introduced into the predicted –10 box (see **Supplementary file 1e** for details). (*Bottom*) Northern blot analysis of pre-CJnc180/CJnc190 in sRNA promoter mutant strains (Δ*rnc* background). (-/+): promoter mutant/WT. Probes for the mature sRNAs

*Figure 3 continued on next page*

*Figure 3 continued*

were used (CSO-0189/0185, respectively, for CJnc180/190; **Figure 1A**). RnpB served as loading control (probed with CSO-0497). For primer extension analysis of the same strains, see panel C.

The online version of this article includes the following source data and figure supplement(s) for figure 3:

**Source data 1.** Full northern blot and primer extension images for the corresponding detail sections shown in **Figure 3**, as well as raw values for western blot quantifications.

**Figure supplement 1.** RNase III affects CJnc190 processing, stability, and *ptmG* regulation.

**Figure supplement 1—source data 1.** Full northern blot images for the corresponding detail sections shown in **Figure 3—figure supplement 1**, as well as raw values for northern blot quantifications.

**Figure supplement 1—source data 2.** Full northern and western blot images for the corresponding detail sections shown in **Figure 3—figure supplement 1**.

**Figure supplement 2.** Northern blot analysis of mature/precursor CJnc180 and CJnc190.

**Figure supplement 2—source data 1.** Full northern blot images for the corresponding detail sections shown in **Figure 3—figure supplement 2**.

**Figure supplement 3.** Mapping of CJnc190 mature and precursor 5′ ends in promoter mutants.

**Figure supplement 3—source data 1.** Full primer extension images for the corresponding detail sections shown in **Figure 3—figure supplement 3**.

---

*supplement 1B*). The combined levels of all CJnc190 species were 10-fold lower in Δ*rnc* compared to WT (**Figure 3A**, compare lanes 1 and 4). Rifampicin stability assays showed that the CJnc180 precursor was stabilized in the Δ*rnc* mutant when compared to WT (**Figure 3—figure supplement 1C**), which further validates processing by RNase III. In contrast, while the half-life of mature CJnc190 in the WT strain was >64 min, CJnc190 precursors had half-lives of 2–4 min in the Δ*rnc* mutant. Taken together, these results indicate that RNase III processes both sRNAs and affects *ptmG* regulation, likely by generating a more stable CJnc190 sRNA species.

## CJnc190 precursors are transcribed from two promoters

To understand the unique RNase III-mediated processing of CJnc180 and CJnc190, we next characterized their precursors. Further northern blot analysis of total RNA from Rnc+ and Δ*rnc* strains with diverse probes showed that the 3′ end of CJnc180 is removed by RNase III processing, and that its 5′ end is RNase III-independent (**Figure 3—figure supplement 2A,B**). Additional probing for CJnc190 in WT and Δ*rnc* suggested that they differ in both their 5′ and 3′ ends (**Figure 3—figure supplement 2A,C**). We used primer extension to map the 5′ ends of the Δ*rnc* precursors for both sRNAs. While CJnc180 had a single, RNase III-independent 5′ end that mapped to its TSS (**Figure 1—figure supplement 2A**), we detected two CJnc190 5′ ends in a Δ*rnc* background (**Figure 3C**, lane 5). One end matched the annotated primary TSS for CJnc190 (**Dugar et al., 2013**), which has a putative –10 box (TAATAT) immediately upstream (**Figure 3C**, right). Inspection of the nucleotides upstream of the second detected 5′ end also revealed a near consensus –10 box (TAAAAT). This suggested that CJnc190 precursors might be transcribed from at least two σ$^{70}$-dependent promoters.

To validate the activity of the three putative CJnc180/190 promoters (annotated CJnc180(P1) and CJnc190(P1); predicted CJnc190(P2)) in vivo, we performed site-directed mutagenesis of their respective –10 boxes in the C-180/190 complementation construct (**Figure 3D**, for details see Materials and methods and **Supplementary file 1e**). Analysis of CJnc180 in a Δ*rnc* background in the generated promoter mutant strains showed that disruption of the annotated CJnc180 promoter –10 box (strain C-190 only) fully abolished CJnc180 expression (**Figure 3D**, lane 5), confirming its single TSS. CJnc180 promoter disruption reduced accumulation of some CJnc190 precursors but did not affect their 5′ ends, suggesting transcriptional interference might impact 3′ ends. Next, we inspected pre-CJnc190 expression in strains with disruptions in either CJnc190(P1) or CJnc190(P2) added to the C-190-only construct. In a strain with CJnc180 and CJnc190 P1 promoters inactivated, the two longest CJnc190 species of Δ*rnc* were absent, and detection of a shorter CJnc190 transcript supported the presence of a second promoter (**Figure 3D**, lane 6). In the strain with only CJnc190 P1 intact, instead only a longer (~280 nt) transcript was detected (lane 7). Finally, disruption of all three putative promoters (C-3×mut) completely abolished expression of all CJnc180 and CJnc190 transcripts (lane 8). Primer extension analysis of RNA from the same promoter mutant strains validated that the two 5′ ends we detected were dependent on the upstream –10 sequences (**Figure 3C**, lanes 7–11). Overall, mapping of CJnc180 and CJnc190 precursors in Δ*rnc* suggested that mature CJnc180 is transcribed from a

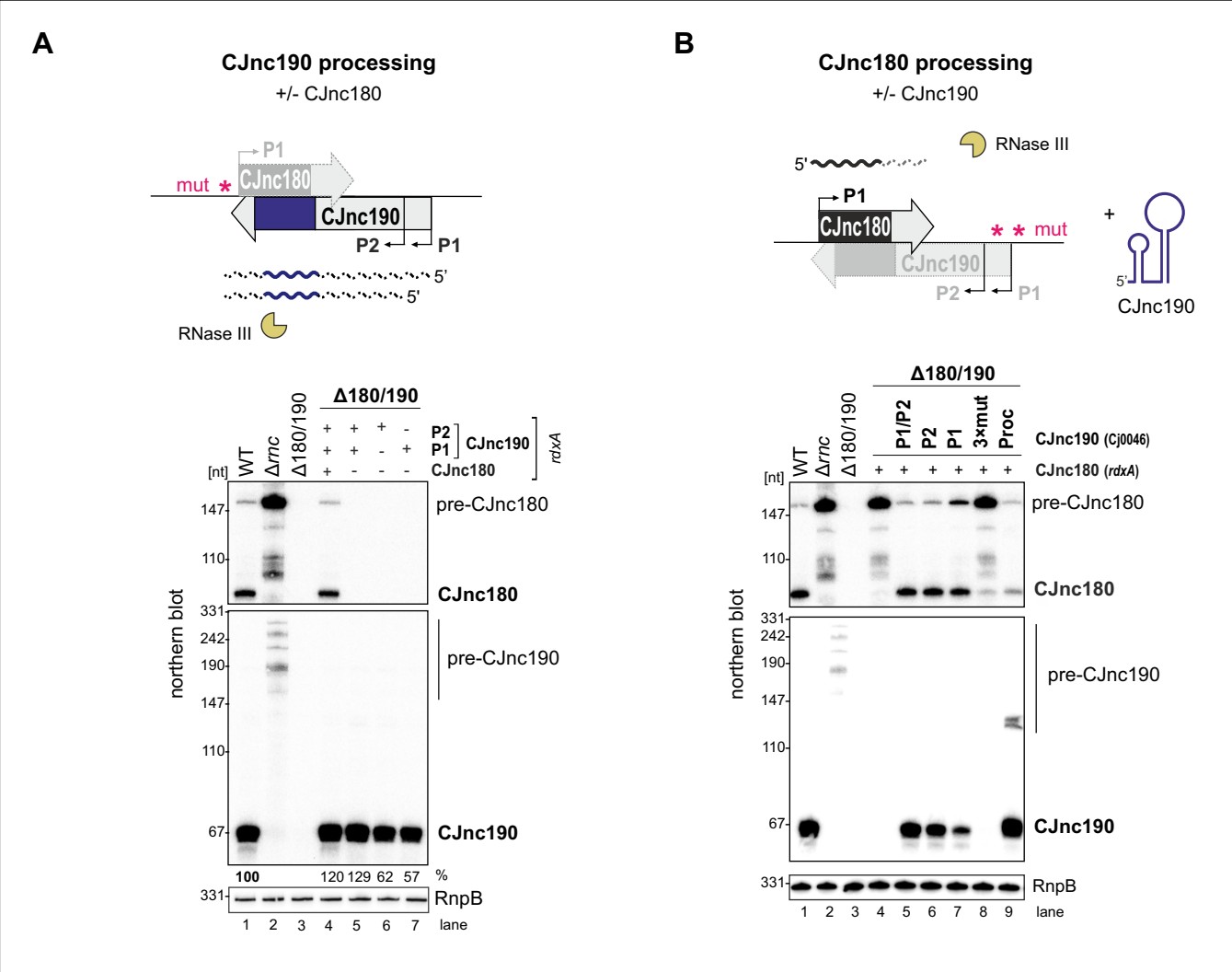

**Figure 4.** CJnc180 requires its antisense partner for RNase III-mediated processing, while CJnc190 processing is CJnc180-independent. Top of both panels: Approach for testing processing in the presence/absence of the antisense partner using promoter mutant alleles. 'mut' – several point mutations introduced into the –10 box (see ***Supplementary file 1e*** for details). (**A**) Northern blot analysis of pre-CJnc190 processing in vivo in the presence/absence CJnc180. The Δ180/190 strain was complemented at *rdxA* with wild type (WT) or CJnc180/190 promoter mutant alleles in an *rnc+* background. (+/-) indicates if a promoter in the CJnc180/190 allele is WT/mutant. (**B**) Pre-CJnc180 processing in the presence or absence of CJnc190. Pre-CJnc180 was introduced into *rdxA* of a Δ180/190 strain. Different CJnc190 species were expressed from the unrelated Cj0046 pseudogene locus. For northern blot detection of CJnc180 and CJnc190, probes for the mature sRNAs were used (CSO-0189 and CSO-0185, respectively). RnpB (probed with CSO-0497) served as a loading control.

The online version of this article includes the following figure supplement(s) for figure 4:

**Source data 1.** Full northern blot images for the corresponding detail sections shown in ***Figure 4***, as well as raw values for northern blot quantifications.

single promoter and derived from its precursor 5' end, while mature CJnc190 is generated from the middle of transcripts with different 3' end lengths arising from two promoters.

## CJnc180, but not CJnc190, requires its antisense partner for processing by RNase III

As the RNase III-dependent biosynthesis pathway of CJnc190 and CJnc180 makes them distinct from processed sRNAs characterized in Gammaproteobacteria, we next explored their maturation in more detail. Because of their extensive complementarity, we hypothesized that RNase III co-processes CJnc180:CJnc190 duplexes. To examine this, we repeated northern blot analysis of the CJnc180/190 promoter-inactivated allele strains, but this time in an RNase III + background. This surprisingly

showed that the C-190-only strain (with the CJnc180 promoter disrupted) still expressed mature CJnc190 (*Figure 4A*). We also found that CJnc190 precursors from either P1 or P2 could give rise to mature CJnc190, although when expressed from P1 or P2 alone, mature sRNA levels were approximately 60% of those detected in the WT, C-180/190, or C-190-only strains. This suggests that both promoters drive transcription of pre-CJnc190 precursors and their combined activity in exponential phase contributes to levels of the mature sRNA.

We next examined whether CJnc180 processing by RNase III is likewise CJnc190-independent. Complementation of Δ180/190 with pre-CJnc180 alone (strain C-180; both CJnc190 promoters disrupted) surprisingly revealed that in contrast to CJnc190, CJnc180 was not processed without its antisense partner (*Figure 4B*, lane 4). Instead, as in Δ*rnc*, we detected only pre-CJnc180. To confirm that CJnc180 processing requires pairing with CJnc190, we added back different CJnc190 species to the second unrelated Cj0046 pseudogene locus in the C-180-only strain (lanes 5–7). Expression of CJnc190 (either P1 or P2) from this second locus restored processing of CJnc180, and even in *trans* expression of 'mature' CJnc190 in the C-190(Proc) strain (which overlaps the CJnc180 3′ end) was sufficient to restore CJnc180 processing (lane 9). Overall, these mutational analyses indicate that while CJnc190 is processed independently of CJnc180, CJnc180 processing requires expression of CJnc190.

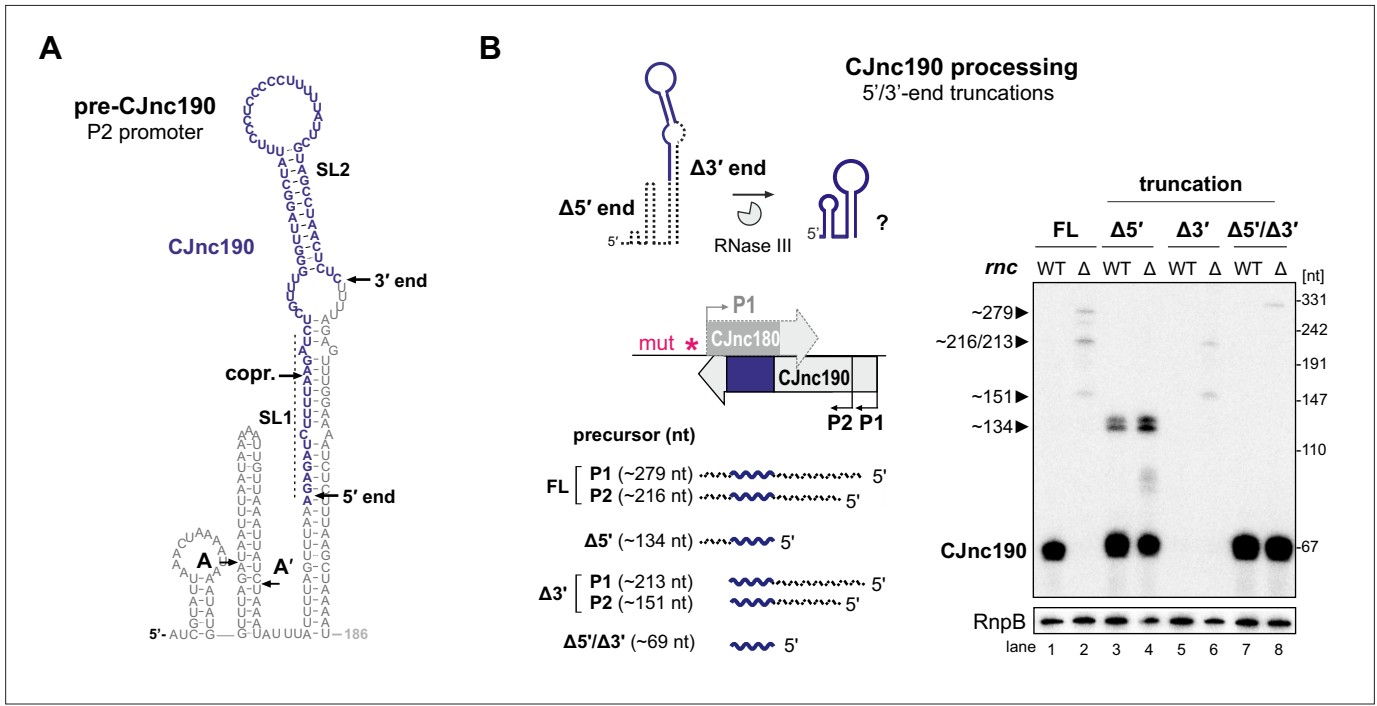

**Figure 5.** Intramolecular duplex-mediated processing of CJnc190. (**A**) Predicted secondary structure of the 186 nucleotide (nt) pre-CJnc190 precursor transcribed from P2. Blue residues: mature sRNA. A/A': putative intermediate 5′ ends identified by primer extension (*Figure 3—figure supplement 3B*). A putative co-processing site ("copr.") with CJnc180 (see *Figure 6A*, *bottom*) is indicated. (**B**) Both 5′ and 3′ ends of CJnc190 are required for processing in vivo. (*Bottom left*) CJnc190 (without CJnc180) was expressed from the *rdxA* locus in Δ180/190 as full-length (FL) or as three versions with truncations at the mature sRNA ends (Δ5′, Δ3′). For 5′ truncations, CJnc190 was fused to its P1 promoter. 'mut' – several point mutations introduced into the predicted CJnc180 –10 box (see *Supplementary file 1e* for details). (*Right*) CJnc190 expression and processing was detected by northern blotting with a probe for the mature sRNA (CSO-0185), while RnpB (CSO-0497) served as a loading control. Total RNA from strains expressing CJnc190 versions on the left were analyzed in an Rnc+ (WT) or Δ*rnc* background. The expected size of each unprocessed precursor is indicated on the left.

The online version of this article includes the following figure supplement(s) for figure 5:

**Source data 1.** Full northern blot images for the corresponding detail sections shown in *Figure 5*.

**Figure supplement 1.** Predicted secondary structures of diverse pre-CJnc190 species detected in vivo.

**Figure supplement 2.** Processing of pre-CJnc190 truncated within a 5′-end hairpin.

**Figure supplement 2—source data 1.** Full northern blot images for the corresponding detail sections shown in *Figure 5—figure supplement 2*.

## CJnc190 processing is mediated by an intramolecular duplex

We hypothesized that CJnc190 processing by RNase III might be mediated by (A) a second *trans*-encoded RNA or (B) via cleavage of an intramolecular duplex. Consistent with the second hypothesis, we detected CJnc190 with a probe designed for a position downstream of its mature 3′ end by northern blotting (*Figure 3—figure supplement 2*), suggesting that both ends might participate in a duplex that is processed by RNase III. We next mapped pre-CJnc190 3′ ends in Δ*rnc* by 3′RACE. The combined northern blot, primer extension, and 3′ RACE information suggested that the most abundant CJnc190 precursor species (~186 nt) arises from P2 with a 3′ extension beyond mature CJnc190 (*Figure 1—figure supplement 3* and *Figure 3—figure supplement 2*). We performed folding predictions for six precursors supported by our 5′ and 3′ mapping (*Figure 5—figure supplement 1*). The most abundant CJnc190 precursor (pre-CJnc190, 186 nt from P2) can fold into a long duplex flanking the mature sRNA, with an additional 5′ hairpin (*Figure 5A*, *left*). While we also detected RNase III cleavage sites within the 5′ hairpin by primer extension (A/A′ in *Figure 5A*; *Figure 3—figure supplement 3B*), deletion of this region did not affect maturation of CJnc190 (*Figure 5—figure supplement 2*). Moreover, these positions do not reflect the mature CJnc190 5′ end. This indicates that the 5′ hairpin of pre-CJnc190 species is not required for processing.

We therefore next examined the requirement of the extended 5′–3′ end duplex flanking the mature CJnc190 sRNA for processing. To test if the 3′ end of the CJnc190 precursor is required for its

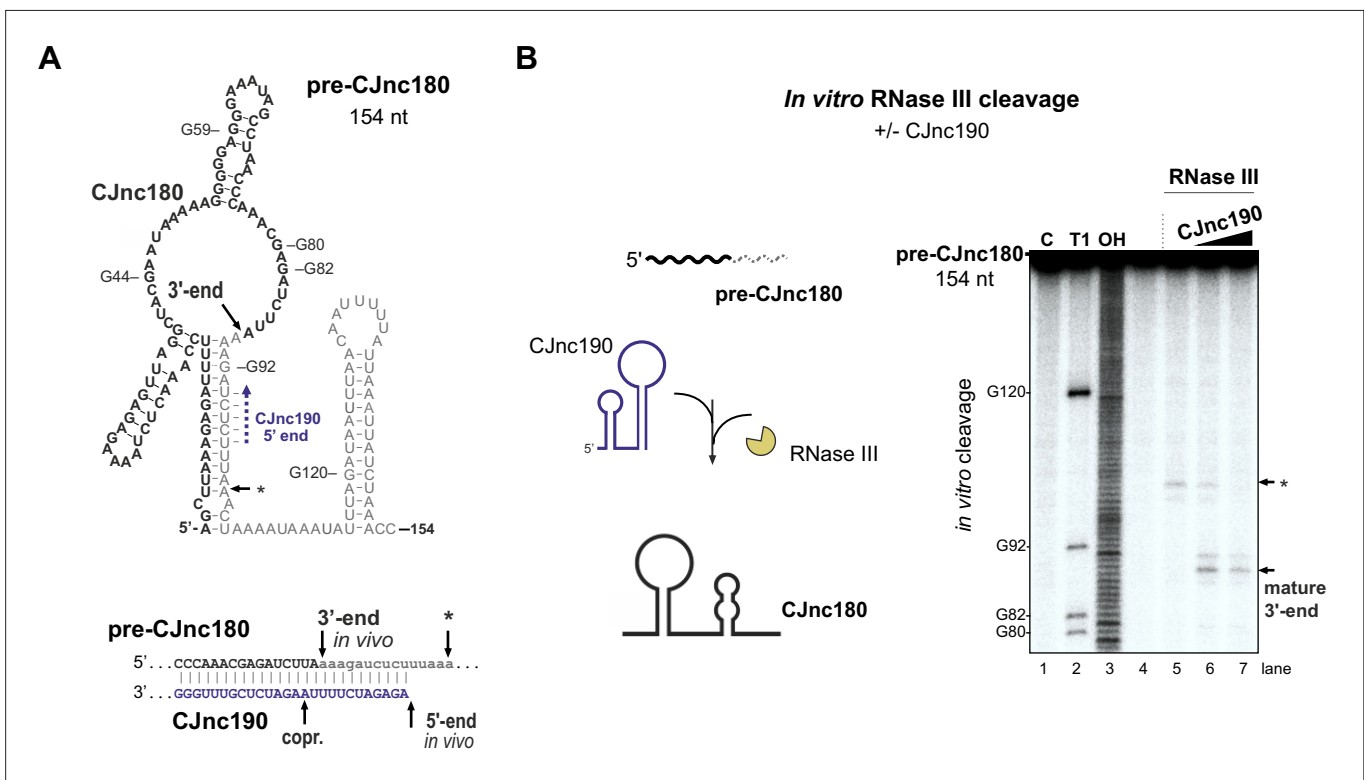

**Figure 6.** CJnc190 drives processing of CJnc180 by RNase III. (**A**) Predicted secondary structure of the pre-CJnc180 precursor (154 nucleotide [nt]). Residues of the mature sRNA are black. The CJnc190-dependent CJnc180 3′ end is indicated. Asterisk: CJnc190-independent in vitro cleavage site (panel B). Blue dashed arrow: Region of potential base-pairing with the 5′ end of mature CJnc190. (*Bottom*) Potential co-processing of the predicted CJnc190 (mature) and pre-CJnc180 duplex. (**B**) In vitro RNase III cleavage of 32P-labeled (5′ end) pre-CJnc180 (154 nt). (*Left*) 32P-5′-end-labeled in vitro transcript (0.2 pmol) was incubated in the presence or absence of unlabeled mature CJnc190 (0.2 or 2 pmol) and subjected to cleavage with RNase III. (*Right*) Cleavage products were separated on a denaturing gel. C – untreated control; T1 ladder – G residues (indicated on left); OH – all positions (alkaline hydrolysis). The full gel is shown in *Figure 6—figure supplement 1B*.

The online version of this article includes the following figure supplement(s) for figure 6:

**Source data 1.** Full cleavage assay gel image for the corresponding detail sections shown in *Figure 6*.

**Figure supplement 1.** In vitro RNase III cleavage of pre-CJnc180.

**Figure supplement 1—source data 1.** Full cleavage assay gel image for the corresponding detail sections shown in *Figure 6—figure supplement 1*.

processing by RNase III, we truncated pre-CJnc190 to the position of the mature sRNA (Δ3′). Unlike a truncation to the mature CJnc190 5′ end, this 3′ end truncation abolished detection of mature CJnc190 (*Figure 5B*, lanes 3/4 and 5/6). Removal of the same 3′ region also abolished processing of either of the 5′-hairpin-truncated species (A or A′) (*Figure 5—figure supplement 2B*, lanes 5/6 and 9/10). In contrast, the mature sRNA was detected when the 3′ end was removed from the 5′-truncated version (*Figure 5B*, lanes 7 and 8). Only CJnc190 precursors with a 3′ extension beyond the mature sRNA showed base-pairing between 5′ and 3′ precursor ends (*Figure 5—figure supplement 1*). Together, this suggests that RNase III cleaves at both sides of the precursor to generate mature CJnc190, and provides insight into how its processing is independent of CJnc180. However, based on mature CJnc190 ends detected in WT, the processing steps following RNase III cleavage remain to be determined.

## CJnc190 drives processing of CJnc180

We next examined why CJnc180 was not processed without CJnc190. We mapped the 3′ ends of pre-CJnc180 by 3′ RACE to positions corresponding to ~104 and ~ 154 nts (major product), consistent with northern blots (*Figure 3A*, *Figure 1—figure supplement 3*). Compared to pre-CJnc190, the predicted secondary structure of the abundant 154 nt pre-CJnc180 species (*Figure 6A*) or 104 nt pre-CJnc180 (*Figure 6—figure supplement 1A*) do not contain as distinct long duplexes as pre-CJnc190. We next performed in vitro RNase III cleavage assays with T7-transcribed pre-CJnc180 (154 nt) in the absence or presence of mature CJnc190 to see if we could recapitulate processing in vitro. In contrast to in vivo, RNase III could cleave pre-CJnc180 even without CJnc190, although the cleavage site was located ~100 nt from the 5′ end (*Figure 6B*, asterisk), rather than at the RNase III-dependent 3′ end of the 88 nt mature sRNA detected in vivo (*Figure 6A*). In contrast, reactions with increasing amounts of mature CJnc190 generated the in vivo RNase III-dependent 3′ end (*Figure 6B*), as well as a second site (*Figure 6—figure supplement 1*, double asterisk). Together, our data suggest that while RNase III is sufficient for maturation of structured CJnc190, CJnc180 also requires its antisense partner for processing.

## CJnc180 indirectly affects *ptmG* via CJnc190 antagonism

While we found that CJnc180 was not required for CJnc190 processing, several *C. jejuni* and *C. coli* strains express this antisense RNA (*Dugar et al., 2013*; *Riedel et al., 2020*; *Svensson and Sharma, 2021*), suggesting it has a conserved function. Because of its extensive complementarity to CJnc190, as well as its co-processing, we hypothesized that it might serve as a CJnc190 antagonist and indirectly affect *ptmG* regulation. We therefore examined the effect of CJnc180 overexpression on PtmG levels. Overexpression of full-length CJnc180 or 'pre-processed' CJnc180(Proc) increased PtmG-3×FLAG to levels intermediate between Δ180/190 and WT (*Figure 7A*). In WT, the ratio of CJnc190 to all CJnc180 transcripts in log phase was ~25:1 (*Figure 7—figure supplement 1B*). CJnc180 over-expression decreased this ratio to ~10:1. This suggested that the CJnc180 antisense RNA could in fact influence *ptmG* indirectly via an effect on CJnc190. In contrast to its overexpression, abolishing CJnc180 expression in log phase did not significantly affect PtmG-3×FLAG levels when CJnc190 was expressed from both P1 and P2 (*Figure 7B*). However, when we expressed CJnc190 from only a single promoter (P1 or P2), thereby reducing its overall levels, the absence of CJnc180 expression significantly affected *ptmG* protein and mRNA (*Figure 7B*, *Figure 7—figure supplement 1C*). In strains with only a single CJnc190 promoter intact [C-190(P1) or C-190(P2)], target levels were intermediate between WT and Δ180/190, in line with the ~2-fold difference in mature CJnc190 levels in these strains (*Figure 7B*, *Figure 7—figure supplement 1D*).

We then wondered when CJnc180 antagonism might come into play. Specific conditions or factors regulating the sRNA pair are not yet known. However, examination of CJnc180 and CJnc190 levels in WT over growth showed that while mature CJnc190 levels increase, levels of CJnc180 fall or remain constant (precursor or mature, respectively) and *ptmG* mRNA levels decrease (*Figure 7C*). We determined the effect of CJnc180 absence/presence on PtmG-3×FLAG levels at different phases of growth (early exponential, mid-exponential, and early stationary). We compared PtmG-3×FLAG protein and mRNA levels in Δ180/190, C-190-only, and OE-180(Proc) strains to those in C-180/190 as a control. PtmG-3×FLAG protein levels were relatively similar in the C-180/190 complemented strain at all culture densities, and in Δ180/190 showed sustained upregulation (*Figure 7D*). In contrast, CJnc180 absence

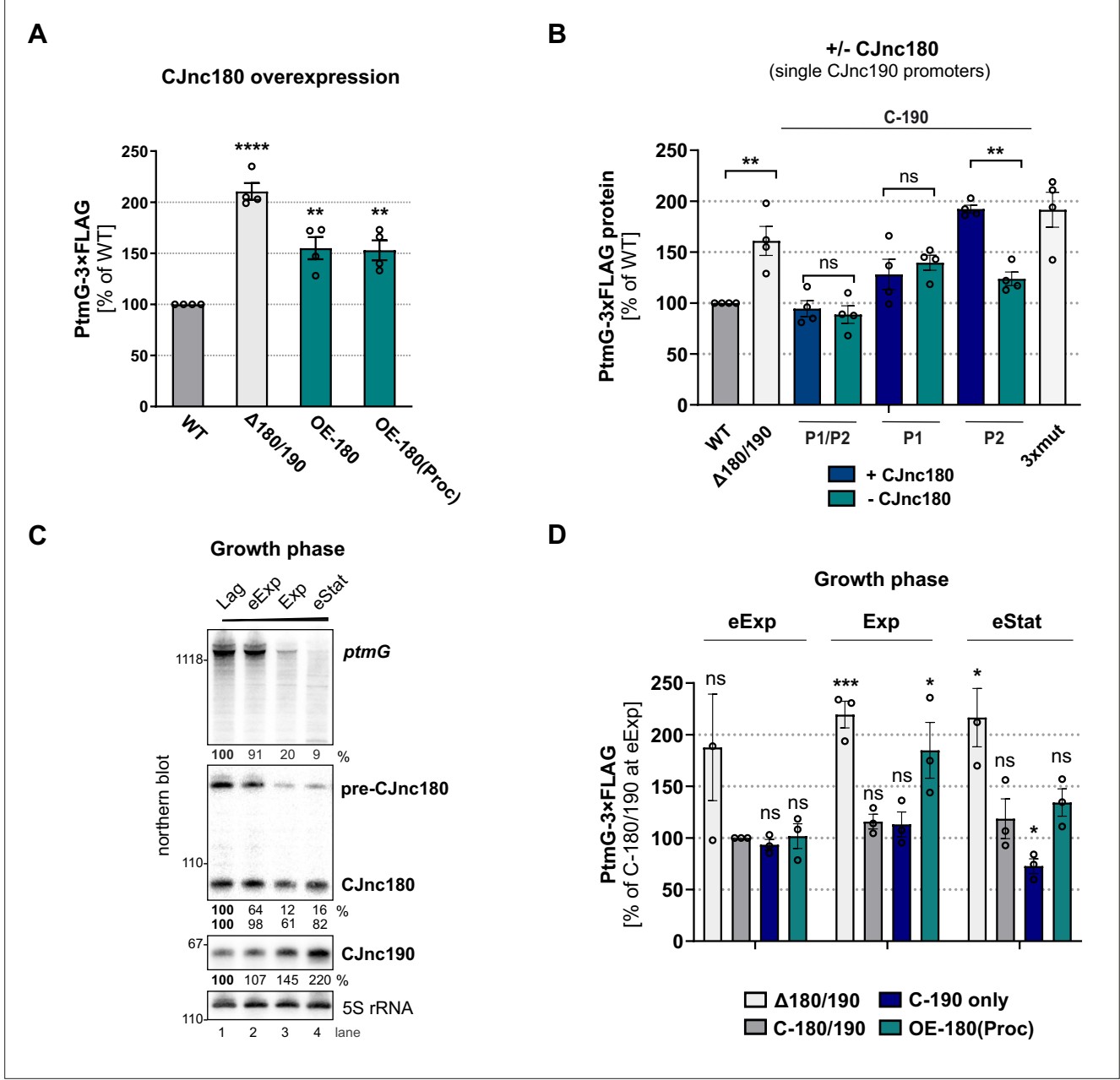

**Figure 7.** CJnc180 antagonizes CJnc190-mediated repression of *ptmG*. (**A**) The effect of CJnc180 overexpression on *ptmG*. OE-180: overexpression (second copy) of full-length CJnc180 from *rdxA*. OE-180(Proc): overexpression (second copy) of mature CJnc180 fused to the *Escherichia coli rrnB* terminator (see *Figure 1D*). Levels of PtmG-3×FLAG protein were measured by western blot in the indicated strains in log phase. Error bars: standard error of the mean (SEM) from four independent replicates. Student's unpaired t-test vs. wild type (WT): **: p < 0.01, ****: p < 0.0001. See also *Figure 7— figure supplement 1A*. (**B**) Absence of CJnc180 derepresses *ptmG* when CJnc190 is expressed from a single promoter. Levels of PtmG-3×FLAG protein were measured by western blot in the indicated strains in log phase. 3× mut: Δ180/190 complemented with CJnc180/190 carrying point mutations in all three validated promoters. Error bars: SEM from four independent replicates. Student's unpaired t-test vs. WT: **: p < 0.01, ns: not significant. See also *Figure 7—figure supplement 1C,D*. (**C**) Northern blot analysis of precursor, mature sRNA, and *ptmG* target mRNA expression in WT at different growth phases in rich medium under microaerobic conditions. Lag: lag phase, eExp: early exponential, Exp: exponential phase, eStat: early stationary phase (OD$_{600}$ 0.1, 0.25, 0.5, and 0.9, respectively). (**D**) The effect of the CJnc180 antagonist at different growth phases. Levels of PtmG-3×FLAG protein were measured in the indicated strains at three growth phases by western blot. Error bars: SEM from three independent replicates. Student's unpaired t-test vs. WT: ***: p < 0.001, *: p < 0.05, ns: not significant vs. C-180/190 (dark gray bars) in eExp. See also *Figure 7—figure supplement 2*. For all northern blots, probes for the mature sRNAs (CSO-0189 and CSO-0185 for CJnc180 and CJnc190, respectively) and the 5′ end of the *ptmG* ORF (CSO-1666) were used. As a loading control, 5S rRNA (CSO-0192) or RnpB (CSO-0497) was also probed.

The online version of this article includes the following figure supplement(s) for figure 7:

*Figure 7 continued on next page*

*Figure 7 continued*

**Source data 1.** Full northern blot images for the corresponding detail sections shown in *Figure 7*, as well as raw values for northern blot quantifications.

**Figure supplement 1.** Modulation of *ptmG* regulation via CJnc180 expression and different CJnc190 promoters.

**Figure supplement 1—source data 1.** Full western and northern blot images for the corresponding detail sections shown in *Figure 7—figure supplement 1*, as well as raw values for northern blot quantifications.

**Figure supplement 2.** Modulation of *ptmG* regulation via CJnc180 expression at different growth phases.

**Figure supplement 2—source data 1.** Full western and northern blot images for the corresponding detail sections shown in *Figure 7—figure supplement 2*, as well as raw values for northern blot quantifications.

**Figure supplement 3.** CJnc180 and CJnc190 levels and promoter activity at different growth phases.

**Figure supplement 3—source data 1.** Full western and northern blot images for the corresponding detail sections shown in *Figure 7—figure supplement 3*, as well as raw values for northern blot quantifications.

**Figure supplement 3—source data 2.** Full western and northern blot images for the corresponding detail sections shown in *Figure 7—figure supplement 3*.

---

or overexpression differentially affected PtmG-3×FLAG levels depending on growth phase. For C-190-only (CJnc180 absent, CJnc190 P1 and P2 active), PtmG-3×FLAG levels were mildly decreased compared to C-180/190 - but only in early stationary. In contrast, overexpression of CJnc180 de-repressed PtmG-3×FLAG levels in mid-log, but had no significant effect in stationary phase. Together, these experiments indicate that CJnc180 acts as a *cis*-acting sRNA antagonist of CJnc190, and that changing the ratios of the two sRNAs determines the outcome of *ptmG* regulation. Further analysis of CJnc190 expression from either promoter over growth did not reveal marked differences in expression (*Figure 7—figure supplement 3A,B*). Moreover, we did not detect strong regulation of CJnc180 or CJnc190 promoter activity over growth (*Figure 7—figure supplement 3C*). Therefore, the signals controlling the levels of the two sRNAs remain to be identified.

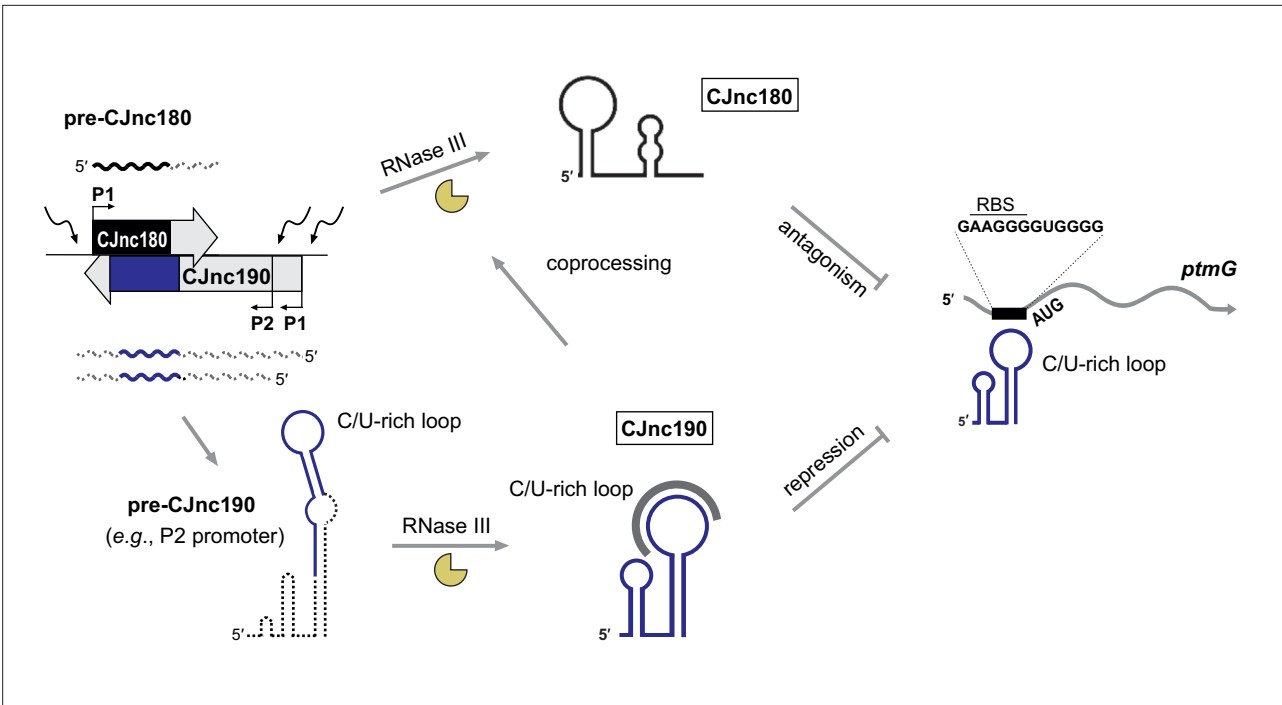

**Figure 8.** CJnc180/190 biogenesis, interplay, and regulation of *ptmG*. (*Left*) CJnc180/CJnc190 precursors are transcribed from one/two promoters, respectively, in response to so far still unknown signals/regulators. (*Bottom*) CJnc190 precursors harbor a long duplex structure involving regions flanking the mature CJnc190 sRNA (blue). Mature CJnc190 is processed from the extended duplex structure in pre-CJnc190 by RNase III in the absence of CJnc180. In contrast, processing of pre-CJnc180 requires both RNase III and duplex formation with CJnc190 (*top*). Mature CJnc190 represses translation of *ptmG* mRNA, encoding a colonization factor, by base-pairing between its C/U-rich loop and the G-rich *ptmG* RBS (*right*). Antisense CJnc180 antagonizes CJnc190 levels/activity by sequestration, decay, and/or transcriptional interference.

## Discussion

Our study of the CJnc190/CJnc180 antisense sRNA pair provides insight into post-transcriptional regulation in the food-borne pathogen *C. jejuni*, as well as more generally into the complex cross-talk among RNA molecules and the role of RNase III in sRNA biogenesis. We have shown that the sRNAs from the virulence-associated CJnc180/190 locus of *C. jejuni* (*Alzheimer et al., 2020*) are processed in a complex biogenesis pathway involving RNase III (summarized in *Figure 8*). We demonstrated that one of these sRNAs, CJnc190, acts as a direct post-transcriptional repressor of the mRNA of the flagellin modification factor PtmG (*Alzheimer et al., 2020*; *Howard et al., 2009*), and that this regulation is antagonized by the *cis*-encoded CJnc180 sRNA. Although the sRNAs are expressed antisense to each other and RNase III processes both, only CJnc180 processing requires its antisense partner, while CJnc190 is processed independently of CJnc180.

### Role of RNase III in sRNA biogenesis

While many enterobacterial sRNAs are generated or activated via processing by RNase E (*Chao et al., 2017*), our study revealed RNase III as a crucial factor for CJnc190 sRNA biogenesis in *C. jejuni*. Only a handful of RNase III-processed bacterial sRNAs have been described so far (*Faubladier et al., 1990*; *Lalaouna et al., 2019*). Our data establishes a role for RNase III in CJnc190 maturation, which in turn is required for *ptmG* regulation. However, it remains unclear why CJnc190 has this complex biogenesis pathway. Processing seems to generate a more stable form of CJnc190, but might also affect its activity. Surprisingly, CJnc190 processing was independent of its antisense RNA CJnc180. Instead, CJnc190 maturation involves co-processing on both sides of the mature sRNA by RNase III via a predicted long duplex region involving both ends of the precursor.

It remains unclear how cleavage of the long CJnc190 5′/3′ end duplex might give rise to the final mature sRNA. Additional RNases such as RNase J, Y, R, or PNPase might be involved in further processing or trimming to the mature 5′/3′ ends. For example, in *E. coli*, processing of ribosomal RNAs or prophage-encoded DicF is initiated by RNase III cleavage of a stem-loop, followed by additional cleavages by RNases such as RNase E (*Faubladier et al., 1990*). RNase III cleavage is followed by RNase J1 trimming during processing of the SRP RNA component scRNA (small cytoplasmic RNA) in *B. subtilis* (*Yao et al., 2007*).

In contrast to CJnc190, CJnc180 processing by RNase III required its *cis*-encoded partner to create a double-stranded substrate. The toxin mRNAs of several type I toxin-antitoxin (TA) systems have long been known to be processed by RNase III upon interaction with their antitoxin RNAs (*Gerdes et al., 1992*; *Vogel et al., 2004*), and in some bacteria such as *Bacillus* this even underlies RNase III essentiality (*Durand et al., 2012*). Besides processing of type I TA loci, RNase III has been implicated in degradation and processing of diverse sense/antisense RNA pairs, including those encoded on plasmids (*Blomberg et al., 1990*), antisense RNA-mRNA duplexes (*Opdyke et al., 2011*), as well as ribosomal RNA precursors (*Iost et al., 2019*), sRNA-mRNA target pairs (*Afonyushkin et al., 2005*; *Romilly et al., 2012*; *Viegas et al., 2011*), and CRISPR/tracrRNA co-processing in Cas9-based CRISPR/Cas systems (*Deltcheva et al., 2011*; *Dugar et al., 2013*).

### CJnc190 is antagonized by the *cis*-encoded CJnc180 sRNA

While CJnc180 is dispensable for CJnc190 processing and *ptmG* repression, we have shown in vivo that CJnc180 is a *cis*-acting antagonist of CJnc190, which can affect *ptmG* regulation. CJnc180 therefore appears to be a new *cis*-acting representative of RNAs that target other sRNAs (reviewed in *Denham, 2020*; *Figueroa-Bossi and Bossi, 2018*; *Grüll and Massé, 2019*). Such antagonists are mainly *trans*-encoded and can be derived from mRNAs, tRNA 3′ external transcribed spacers, or other sRNAs. In contrast to reported examples, CJnc180 antagonizes a *cis*-encoded sRNA. Additional candidate antagonizing RNAs are *cis*-encoded pairs that are differentially expressed under specific conditions (*Denham, 2020*). For example, antisense SraC/SdsR (RyeA/RyeB), widely conserved in *E. coli* and *Salmonella* (*Fröhlich et al., 2016*), show reciprocal expression and are also processed by RNase III (*Vogel et al., 2003*). In *E. coli*, SdsR overexpression leads to cell death via repression of *yhcB*, which is rescued by SraC overexpression, and the pair was proposed to be a novel TA system where both components are sRNAs (*Choi et al., 2019*; *Choi et al., 2018*; *Gupta et al., 2019*). Whether the *cis*-acting antagonist CJnc180 from the CJnc190/180 locus also has an additional function and can act on other RNAs in *trans* remains to be seen.

While the exact mechanism of antagonism is unclear, hypotheses can be made based on other *cis*-acting sRNAs (*Brantl, 2007*). An open question is whether the two sRNAs simply sequester each other, or play a role in each others' turnover. CJnc180:CJnc190 co-processing would disrupt SL1 of CJnc190, which potentially protects the sRNA from RNase-mediated degradation. Antagonism could thus occur via cleavage and decay of CJnc190. Inspection of CJnc190 primer extension analysis for evidence of co-processing of the two sRNAs in vivo (*Figure 3—figure supplement 3B*) did not reveal an RNase III- and CJnc180-dependent CJnc190 5′ end that would be consistent with the 2 nt 3′-overhangs generated by RNase III (*Figure 6A*, bottom), although reduced stability might preclude its detection. Alternatively, only the CJnc180 strand of the duplex might be cleaved, as has been reported for some RNase III substrates (*Altuvia et al., 2018*; *Court et al., 2013*; *Le Rhun et al., 2017*; *Dunn, 1976*).

Since overexpression of either 'pre-processed' or full-length CJnc180 affected *ptmG*, co-processing might not be the only mechanism by which CJnc180 influences CJnc190. The extensive complementarity remaining for the processed sRNAs means that even mature CJnc180 could sequester CJnc190 – and possibly also promote degradation by RNase III. Finally, transcriptional interference might be involved, in line with our observation that abolishing CJnc180 expression reduced levels of some CJnc190 precursors with shorter 3′ ends (*Figure 3D*). The CJnc190 3′ end position is immediately upstream of the CJnc180 promoter on the opposite strand (*Figure 1—figure supplement 3*). RNAs (including sRNAs, mRNAs, or derivatives of other cellular RNAs) that target sRNAs have commonly been termed 'sponge' RNAs or ceRNAs (*Denham, 2020*; *Grüll and Massé, 2019*). Ultimately, the impact of CJnc180 on CJnc190 (and vice versa) might be multifactorial and include sequestration from targets, transcriptional interference, or promotion of decay. This might depend on their relative levels and/or on levels of other cellular molecules such as mRNA targets and RNases, as post-transcripional regulation is inherently dependent on the overall cellular state (*Gottesman, 2004*). We therefore propose terming CJnc180 (and also CJnc190) as RNA 'antagonists', and suggest reserving the term 'sponge' for those that purely sequester or compete with targets without promoting decay.

The potential for independent regulation of CJnc180 and CJnc190 makes an antagonistic relationship attractive. The mechanism by which CJnc180 affects CJnc190 appears to be dependent on the stoichiometry of the two sRNAs, as has been previously proposed (*Denham, 2020*) and recently demonstrated for the RNAIII antagonist SprY (*Le Huyen et al., 2021*). A similar variation in sponge and sRNA stoichiometry has been reported for the ChiX sRNA and its decoy, the *chbBC* intergenic region (*Figueroa-Bossi et al., 2009*; *Plumbridge et al., 2014*). Presumably, CJnc180 could serve to buffer and/or set a threshold for CJnc190 levels to regulate targets, as for tRNA-derived sponge RNAs in *E. coli* (*Lalaouna et al., 2015*). While we observed that the mature sRNAs show inverse expression levels during growth, it remains unknown how they are transcriptionally controlled and in response to which signals their levels change. While CJnc180 is transcribed from one promoter (P1), CJnc190 is transcribed from at least two promoters (P1 and P2). Precursors from both promoters are expressed during routine growth and give rise to the mature sRNA. The two promoters presumably increase the potential for environmental inputs into CJnc180/190, thereby increasing the complexity of the locus even further. Future work will reveal transcriptional regulators and conditions that regulate the CJnc180/190 promoters, and how transcriptional control intersects with RNase III processing.

## CJnc190 directly represses *ptmG* by targeting a G-rich sequence and impacts virulence

Based on in vitro and in vivo analyses, we demonstrated that CJnc190 directly represses *ptmG* translation by base-pairing with the G-rich RBS of its mRNA using a C/U-rich loop. Thus, CJnc190 binding likely interferes with translation initiation, resembling the canonical mode of sRNA-mediated target repression (*Storz et al., 2011*). The structure of mature CJnc190 is reminiscent of RepG sRNA from *Helicobacter pylori* (*Svensson and Sharma, 2021*), which uses a C/U-rich loop to target a phase-variable G-repeat in the 5′ UTR of a chemotaxis receptor mRNA (*Pernitzsch et al., 2014*). Although CJnc190 and RepG share C/U-rich loop regions, their biogenesis is strikingly different: while RepG is transcribed as a separate standing gene corresponding to the mature sRNA (*Pernitzsch et al., 2014*), CJnc190 is transcribed opposite to another sRNA (CJnc180) and is processed into the mature form from diverse precursors in *C. jejuni* and related *C. coli* (*Svensson and Sharma, 2021*). It is interesting to imagine how CJnc190 (and CJnc180) might have arisen - de novo (*Jose et al., 2019*, *Updegrove*

*et al., 2015*), or from a degenerate type I toxin-antitoxin system as might be the case for the *H. pylori* sRNA NikS (*Eisenbart et al., 2020*). Also in more diverse species, several sRNAs with C/U-rich target interaction sites have been reported (*Boisset et al., 2007*; *Bronesky et al., 2016*; *Heidrich et al., 2017*; *Schmidtke et al., 2013*), suggesting that targeting of G-rich sequences might be a more widespread phenomenon.

A previous study of CJnc180/190 in our 3D model of the human intestine (*Alzheimer et al., 2020*) showed that CJnc180/190 as well as *ptmG*, the target of CJnc190, are involved in infection. PtmG is part of a six-gene cluster in the flagellin glycosylation island of *C. jejuni* NCTC11168 that has been associated with livestock strains (*Champion et al., 2005*) and is in the pathway generating legion-aminic acid sugar precursors that decorate its flagellins (*Howard et al., 2009*; *Zebian et al., 2016*). As PtmG is required for chicken colonization (*Howard et al., 2009*), CJnc190 might impact virulence phenotypes via *ptmG* regulation. However, as CJnc190 is also found in strains that lack PtmG, such as strain 81–176 (*Dugar et al., 2013*), it likely has additional targets that remain to be identified that could account for its infection phenotype (*Alzheimer et al., 2020*). In addition, antagonist CJnc180, which indirectly affects *ptmG* levels, might have an additional function as a *trans*-acting sRNA that targets other mRNAs encoding factors affecting virulence. Future studies will reveal the complete regulon of each sRNA and the contribution of their direct targets to virulence phenotypes.

## Materials and methods

**Key resources table**

| Reagent type (species) or resource | Designation | Source or reference | Identifiers | Additional information |
|---|---|---|---|---|
| Strain, strain background | *Escherichia coli* TOP10 | Invitrogen | | See *Supplementary file 1b* |
| Strain, strain background | *Campylobacter jejuni* NCTC11168 | Arnoud van Vliet, Institute of Food Research, Norwich, UK | | See *Supplementary file 1b* |
| Sequence-based reagent | (Oligonucleotides) See *Supplementary file 1c* | Sigma | | See *Supplementary file 1c* |
| Recombinant DNA reagent | (Plasmids) See *Supplementary file 1d* | This study | | See *Supplementary file 1d* |
| Antibody | Anti-FLAG M2 (mouse monoclonal) | Sigma-Aldrich | #F1804-1MG | Western blot 1:10,000 |
| Antibody | Anti-GFP (mouse monoclonal) | Roche | #11814460001 | Western blot 1:1000 |
| Antibody | Anti-GroEL (rabbit polyclonal) | Sigma-Aldrich | #G6532-5ML | 1:10,000 |
| Antibody | Anti-mouse HRP conjugate IgG (sheep polyclonal) | GE Healthcare | #RPN4201 | 1:10,000 |
| Antibody | Anti-rabbit HRP conjugate IgG (goat polyclonal) | GE Healthcare | #RPN4301 | 1:10,000 |
| Commercial assay or kit | PURExpress | New England Biolabs | E6800S | |
| Commercial assay or kit | DNA Cycle Sequencing Kit | Jena Bioscience | #PCR-401S | |
| Commercial assay or kit | MEGAscript T7 kit (Ambion) | Thermo Fisher Scientific | AMB13345 | |
| Software, algorithm | RNAfold | *Lorenz et al., 2011* (PMID:22115189) | Vienna RNA package 2.4.14 | http://www.tbi.univie.ac.at/RNA |
| Software, algorithm | IntaRNA | *Mann et al., 2017* (PMID:28472523) | Version 3.2.0 linking Vienna RNA package 2.4.14 | http://rna.informatik.uni-freiburg.de/IntaRNA/Input.jsp |
| Software, algorithm | AIDA | Raytest, Germany | Version 5.0 SP1 Build 1,182 | |
| Software, algorithm | Integrated Genome Browser | UNC Charlotte | Version 9.1.8 | bioviz.org |

## Bacterial strains and culture conditions

*C. jejuni* strains (*Supplementary file 1b*) were routinely grown either on Müller-Hinton (MH) agar plates or with shaking at 140 rpm in *Brucella* broth (BB) at 37°C in a microaerobic atmosphere (10% $CO_2$, 5% $O_2$). All *C. jejuni* media was supplemented with 10 µg/ml vancomycin. Agar was also supplemented

with marker-selective antibiotics (20 µg/ml chloramphenicol [Cm], 50 µg/ml kanamycin [Kan], 20 µg/ml gentamicin [Gm], or 250 µg/ml hygromycin B [Hyg]) where appropriate. *E. coli* strains were grown aerobically at 37°C in Luria-Bertani (LB) broth or on LB agar supplemented with the appropriate antibiotics for marker selection.

## General recombinant DNA techniques

Oligonucleotide primers for PCR, site-directed mutagenesis, Sanger sequencing, and northern blot probing are listed in *Supplementary file 1c* and were purchased from Sigma. Plasmids generated and/or used in this study are listed in *Supplementary file 1d*. Site-directed mutagenesis was performed on plasmids by inverse PCR with mutagenic primers as listed in *Supplementary file 1e*, according to standard protocols. DNA constructs and mutations were confirmed by Sanger sequencing (Macrogen). Restriction enzymes, *Taq* polymerase for validation PCR, and T4 DNA ligase were purchased from NEB. For cloning purposes, Phusion high-fidelity DNA polymerase (Thermo Fisher Scientific) was used. For PCR amplification of constructs containing the Hyg$^R$ cassette, 3% DMSO was included in reactions.

## Transformation of *C. jejuni* for mutant construction

All *C. jejuni* mutant strains (deletion, chromosomal 3×FLAG-tagging, chromosomal point mutations, listed in *Supplementary file 1b*) were constructed by double-crossover homologous recombination with DNA fragments introduced by electroporation or natural transformation. For electroporation, strains grown from frozen stocks until passage 1 or 2 on MH agar were harvested into cold electroporation solution (272 mM sucrose, 15% (v/v) glycerol) and washed twice with the same buffer. Cells (50 µl) were mixed with 200–400 ng PCR product on ice and electroporated (Bio-Rad MicroPulser) in a 1 mm gap cuvette at 2.5 kV. Cells were then transferred with *Brucella* broth to a non-selective MH plate and recovered overnight at 37°C microaerobically before plating on the appropriate selective medium and incubating until colonies were visible (2–4 days). For natural transformation, approximately 100–1000 ng of genomic DNA isolated from the desired donor strain was mixed with the acceptor strain on non-selective MH plates and incubated for 4–5 hr microaerobically at 37°C. The transformation mixture was then transferred to the appropriate selective medium and incubated until colonies were visible.

## *C. jejuni* deletion mutant construction by recombination with overlap PCR products

Deletion of the CJnc180/190 locus with a polar deletion cassette in *C. jejuni* NCTC11168 has been previously described (*Alzheimer et al., 2020*). Non-polar deletion mutants of protein-coding genes were constructed by homologous recombination with overlap PCR products consisting of a non-polar resistance cassette in between approximately 500 bp of sequence upstream and downstream of the target gene using primer pairs listed in *Supplementary file 1f*. As an example, deletion of *rnc* in *C. jejuni* NCTC11168 with a non-polar Hyg$^R$ cassette is described. The approximately 500 bp region upstream of *rnc* (Cj1635c) was amplified using CSO-0240/0241, while the downstream region was amplified using CSO-0242/0243. The 5' ends of the antisense primer for the upstream region and the sense primer for the downstream region included regions overlapping the resistance cassette 5' and 3' end, respectively. The Hyg$^R$ cassette was amplified using primers CSO-1678/1679 from pACH1 (*Cameron and Gaynor, 2014*). Next, a three-fragment overlap PCR was performed using the *rnc* upstream, *rnc* downstream, and Hyg$^R$ cassette fragments in an equimolar ratio and primers CSO-0240/0243. Following confirmation of the correct size by gel electrophoresis, the resulting overlap PCR product was electroporated as described above into WT. Deletion mutants were selected on plates containing hygromycin. The deletion strain (Δ*rnc*) was confirmed using a primer binding upstream of the *rnc* upstream fragment (CSO-0239) and an antisense primer binding the Hyg$^R$ cassette (CSO-2857). For non-polar Kan$^R$ and Gm$^R$ deletions, cassettes were amplified using HPK1/HPK2 (*Pernitzsch et al., 2014*; *Sharma et al., 2010*) and pGG1 (*Dugar et al., 2016*) or pUC1813-apra (*Bury-Moné et al., 2003*) as a template, respectively.

## C-terminal 3×FLAG-tagging of *ptmG*

A 3×FLAG epitope was fused to the penultimate codon of *ptmG* at its native locus by homologous recombination with an overlap PCR product, which contained the upstream region of *ptmG* and its CDS

to the penultimate codon fused to the 3×FLAG sequence, a Gm$^R$ resistance cassette, and the *ptmG* downstream region. The *ptmG* upstream and coding region was amplified using CSO-1538/1539, and 500 bp of the downstream was amplified with CSO-1540/1541. A fusion of 3×FLAG to the Gm$^R$ cassette was amplified using JVO-5142/HPK2 on the previously published 3×FLAG strain (*fliW*::3×-FLAG) (*Dugar et al., 2016*).

## Heterologous expression from *rdxA*

The *rdxA* locus (Cj1066) can be used for heterologous gene expression in *C. jejuni* (*Ribardo et al., 2010*). Constructs for complementation in *rdxA* were made in plasmids containing approximately 500 bp of upstream and downstream sequence from *rdxA* flanking a Cm$^R$ or Kan$^R$ cassette (with promoter and terminator) by subcloning the *C. jejuni* sequence into previously constructed plasmid vectors based on pST1.1 (*Dugar et al., 2018*) or pGD34.7 (*Alzheimer et al., 2020*). The CJnc180/190 and *ptmG* complementation plasmids (pGD34.7 and pSSv63.1) have been described previously (*Alzheimer et al., 2020*). Primers CSO-2276/2277 were then used to amplify all constructs from *rdxA*-based plasmids for electroporation into *C. jejuni*. Clones with intended insertions were validated by colony PCR using CSO-0643/0349 (*rdxA*-Cm$^R$ constructs) or CSO-0023/0349 (*rdxA*-Kan$^R$ constructs). Insertions were sequenced with CSO-3270, CSO-0643, or CSO-0023.

## Generation of plasmids for expression of processed/truncated (5′ or 3′) CJnc190 or CJnc180 (3′) at *rdxA*

To express "pre-processed" CJnc190 from its native promoter P1, the 5′ end of mature CJnc190 detected by primer extension was removed from pGD34.7 by inverse PCR with primers CSO-2109/1545. After *Dpn*I digestion, ligation, and transformation into *E. coli*, the correct plasmid (pSSv20.1) was verified by colony PCR with pZE-A/pZE-XbaI and sequencing using CSO-0354. Truncations in the 5′ hairpin (sites A and A′) were generated in a similar fashion, using inverse PCR on pSSv80.1 with primers CSO-5235/1545 (A, pSSv148.1) or CSO-5234/1545 (A′, pSSv147.1). Following *Dpn*I digestion, ligation, and transformation into *E. coli* TOP10, positive clones were identified by colony PCR with CSO-0643/3270 and sequenced with CSO-0643. All CJnc190 3′-end truncations were generated by inverse PCR with CSO-0347/5391 on plasmids pSSv20.1, pSSv147.1, pSSv148.1, and pSSv80.1 to generate plasmids pSSv160.2, pSSv162.1, pSSv163.1, and pSSv161.1, respectively. Clones were also identified by colony PCR with CSO-0643/3270 and sequenced with CSO-0643. For the expression of processed CJnc180 with an *E. coli rrnB* terminator, plasmid pGD34.7 was used as template for inverse PCR using the primers CSO-3831/3832. After *Dpn*I digestion, ligation, and transformation into *E. coli*, the correct plasmid (pSSv116.1) was verified by colony PCR with CSO-0643/3270 and sequencing using CSO-3270.

## Generation of a plasmid for expression from the Cj0046 pseudogene locus

The Cj0046 pseudogene locus has been used previously as a *C. jejuni* heterologous expression/complementation locus (*Kim et al., 2008*). To generate a plasmid allowing insertion of genes with a resistance cassette within Cj0046, ~1000 bp of the Cj0046 locus was first amplified using primers CSO-1402/1405. Recipient plasmid pSSv1.2 (CSS-1125; *Dugar et al., 2016*) was amplified by inverse PCR with CSO-0073/0075. The vector and insert were digested with *Xba*I and *Xho*I and ligated together. A positive plasmid clone (pSSv53.1, CSS-2861) was validated by colony PCR with CSO-1402/1405 and sequencing with CSO-1402. Next, pSSv53.1 was amplified by inverse PCR with CSO-2748/2751 and digested with *Eco*RI. A polar (promoter and terminator) Gm$^R$ cassette was amplified with CSO-2749/2750 on pGD46.1 (CSS-0858) and similarly digested. The pSSv53.1 backbone and Gm$^R$ cassette were ligated together to generate pSSv54.3, which was validated by colony PCR with CSO-1402/1405.

## Generation of a plasmid for insertion of CJnc190 at Cj0046

To generate plasmids containing CJnc180/190 alleles with a Gm$^R$ cassette flanked by 500 bp of Cj0046, the WT sRNA region was first amplified from *C. jejuni* NCTC11168 WT (CSS-0032) gDNA with CSO-0354/0355. The pSSv54.3 plasmid (CSS-2872) was then amplified by inverse PCR with CSO-2750/2751. Both insert and vector were digested with *Nde*I and *Cla*I and ligated. A positive clone (pSSv55.4, CSS-2965) was identified by colony PCR with CSO-0833/0354. Point mutant alleles were

generated by site-directed mutagenesis with primer pairs listed in **Supplementary file 1e**. All mutations were validated by sequencing with CSO-0833. For insertion of processed CJnc190 into the Cj0046 plasmid (pSSv56.6), the sRNA region from pSSv20.1 (CSS-1610) was amplified by PCR with CSO-2839/0354 and the pSSv54.3 (CSS-2872) vector was amplified with CSO-2750/2751. Both vector and insert were digested with NdeI and ClaI and ligated. A positive clone was identified by colony PCR with CSO-2839/1405 and sequencing with CSO-2839.

## Construction of *ptmG(10th)*-GFP translational fusions at Cj0046 for point mutant analysis

A *gfpmut3* translational fusion reporter for *ptmG* was constructed as follows, starting with plasmid pST1.1 (**Dugar et al., 2018**). First, the *metK* promoter of pST1.1 was replaced with the promoter, 5′ UTR, and the first 10 codons of *ptmG*. The *ptmG* 5′ UTR and first 10 codons were amplified by PCR using CSO-1670/1671 and digested with NdeI. Plasmid pST1.1 was then amplified by PCR with primers designed to remove the *metK* promoter (CSO-1669/0762) and similarly digested. Ligation of the pST1.1 backbone and *ptmG* region resulted in plasmid pSSv15.1, which was confirmed by colony PCR with CSO-1670/0348 and sequencing with CSO-0023. Next, the *ptmG(10th)*-GFP reporter fusion was moved into the Cj0046 pseudogene insertion plasmid pSSv54.3 as follows. The region of pSSv15.1 harboring the reporter and Kan[R] cassette was amplified using CSO-0159/2877, and pSSv54.3 was amplified using CSO-2818/2870, which removes the Gm[R] cassette. Both insert and vector were digested with PstI and NotI and then ligated to make pSSv61.1, which was validated by colony PCR with CSO-0023/0789. A single G-to-C point mutation (M1′) was introduced into the 5′ UTR of *ptmG* in pSSv61.1 by inverse PCR with the mutagenic primer pair CSO-2875/2876 to create pSSv62.1. The mutation was confirmed by sequencing with CSO-0789. Next, the *rdxA*::Kan[R]-*ptmG*UTR-GFP construct from the WT and M1′ plasmids was amplified using CSO-1402/1405 and transformed into *C. jejuni*. Insertions were confirmed by colony PCR with CSO-0023/3217 and sequencing with CSO-0023. Strains in these backgrounds deleted for the sRNAs and/or complemented at *rdxA* were then constructed as for the WT background, described above, except for the addition of CJnc190 M1 point mutant, which was made by site-directed mutagenesis of pSSv20.1 with CSO-2871/2872 and confirmed by sequencing with CSO-0643.

## Generation of promoter exchange experiment strains

The *ptmG* promoter (P$_{ptmG}$) was exchanged for the *flaA* promoter (P$_{flaA}$) in the above *ptmG(10th)*-GFP translational fusion as follows. Plasmid pSSv61.1 (CSS-2921) was amplified by inverse PCR with CSO-0762/1955 and the *flaA* promoter was amplified from WT gDNA (CSS-0032) with CSO-1732/1733. Both amplicons were digested with XmaI and ligated. A positive clone (pSSv83.1, CSS-3282) was validated by colony PCR with CSO-0023/1933 and sequencing with CSO-0023. The region of interest was amplified from pSSv83.1 with primers CSO-1402/1405 and transformed into *C. jejuni* Δ*ptmG*_UTR (CSS-3234). Positive clones were checked by colony PCR with CSO-0023/0789 to validate strain CSS-3302.

A control fusion of the *flaA* promoter was constructed as follows. First, the intermediate plasmid pGD7.1 (**Alzheimer et al., 2020**) was amplified by inverse PCR with CSO-0482/0493, digested with NotI/PstI, and ligated with a similarly digested polar Kan[R] cassette-P$_{ureA}$-*gfpmut3* PCR product amplified with CSO-0513/0414 from p463 (CSS-0079; **Pernitzsch et al., 2014**) to generate plasmid pMA4.5 (CSS-2389), which was validated by colony PCR with CSO-0348/0513 and sequencing with CSO-0345/0348. The backbone of pMA4.5 was then amplified by inverse PCR with CSO-1734/0762, and the *flaA* (Cj1339c) promoter was amplified from *C. jejuni* NCTC11168 WT (CSS-0032) gDNA with CSO-1732/1733. Both vector and insert were digested with XmaI and EcoRI and ligated to create pST10.1 (CSS-2379), which was validated by colony PCR with CSO-0023/0348 and sequencing with CSO-0348. The reporter regions were amplified with CSO-2276/2277 for electroporation into *C. jejuni*.

The *fliA* gene (Cj0061) was disrupted in the pSSv61.1 (P$_{ptmG}$-*ptmG(10th)*-GFP), pSSv83.1 (P$_{flaA}$-*ptmG(10th)*-GFP), and pST10.1 (P$_{flaA}$-*flaA*-GFP) promoter fusion strains (CSS-2945, CSS-3301, and CSS-3388, respectively) by natural transformation of genomic DNA from a strain carrying the *fliA* region with a non-polar Gm[R] cassette (CSS-1133; **Dugar et al., 2016**) to generate CSS-3983, CSS-3303, and CSS-4025. Deletion of *fliA* was confirmed by colony PCR with CSO-1153/HPK2. The CJnc180/190 region was disrupted in CSS-3301 and CSS-3388 with a polar Gm[R] cassette by transformation with an

overlap PCR product generated as described above and validation by colony PCR with CSO-0246/HPK1 to generate strains CSS-3305 and CSS-3390. The P*flaA*-*ptmG(10th)*-GFP Δ180/190 strain (CSS-3305) was then complemented at *rdxA* with the mature CJnc190 region from pSSv20.1 to generate CSS-2957. For this, a PCR product amplified with CSO-2276/2277 on pSSv20.1 (CSS-1610) was electroporated into CSS-3305 and colonies were checked by colony PCR with CSO-0349/0643 and sequencing with CSO-0643.

## Generation of promoter point mutant alleles

For construction of different CJnc180/190 alleles, the original pGD34.7 or pSSv55.1 plasmids were subjected to site-directed mutagenesis by inverse PCR/*Dpn*I digestion using primers listed in **Supplementary file 1e** to create otherwise isogenic plasmids listed in **Supplementary file 1d**. First, the CJnc180(P1) promoter was disrupted (see **Supplementary file 1e** for mutations) to create strain C-190-only [CJnc190(P1) and CJnc190(P2) active] (plasmid pSSv38). To this allele, CJnc190(P1) or (P2) promoter inactivation mutations were added to create strains C-190(P2) only and C-190(P1) only (pSSv98.1 and pSSv97.1), respectively. Finally, a strain with all three putative promoters mutated (C-3×mut) was also constructed (pSSv79.1). The different *rdxA*::Cm^R-CJnc180/190 alleles were amplified from the plasmids using CSO-2276/2277 and electroporated into the *C. jejuni* Δ180/190 strain (CSS-1157). The strains were validated for the correct insertion of each transformed allele by colony PCR using CSO-0643/0349 and sequencing with CSO-0643.

## Transcriptional fusions to superfolder GFP

The CJnc180 promoter from *C. jejuni* and CJnc180 upstream region from *C. coli* were fused to a promoterless superfolder GFP (sfGFP) cassette with the RBS from *hupB* (Cj0913c) by overlap PCR as follows. An *rdxA*UP(approx. 500 bp)-Kan^R-sfGFP cassette was first generated in a plasmid by amplification of sfGFP from pXG10 (**Corcoran et al., 2012**) with primers CSO-3279/3569, digestion with *Xma*I, and ligation to pST1 (**Dugar et al., 2018**). The pST1 backbone was amplified with CSO-0762/0347 and similarly digested with *Xma*I. This plasmid was validated by colony PCR with CSO-0023/3527 and sequencing with CSO-0023. The *rdxA*UP-Kan^R-sfGFP cassette was then amplified with primers CSO-5590/2276, and the *rdxA*DN region (approx. 500 bp) was amplified with CSO-0347/2277. CJnc180 promoter/upstream regions were amplified with CSO-5595/5593 and CSO-5597/5598 (*C. jejuni* and *C. coli*, respectively) from WT genomic DNA to introduce regions overlapping the *rdxA*DN fragment or *hupB* RBS. The three fragments (*rdxA*DN/promoter/sfGFP-Kan^R-*rdxA*UP) were mixed, annealed, and amplified by overlap PCR with CSO-2276/2277. The resulting PCR product was electroporated into *C. jejuni* NCTC11168 WT. Kan^R colonies were validated for insertion of the transcriptional fusion at *rdxA* by colony PCR with CSO-0349/0789 and promoter regions were checked by sequencing with CSO-3270.

## Total RNA extraction and analysis by northern blotting

For analysis of total RNA, bacterial strains were grown to log phase in BB and approximately 4 $OD_{600}$ were harvested and mixed with 0.2 volumes of stop-mix (95% ethanol and 5% phenol, v/v). Samples were immediately snap-frozen in liquid nitrogen and stored at −80°C until RNA extraction. Frozen samples were thawed on ice and centrifuged at 4°C to collect cell pellets (4500 *g*, 20 min), which were then lysed by resuspension in 600 µl of a solution containing 0.5 mg/ml lysozyme and 1% SDS in Tris-EDTA buffer (pH 8.0) and incubation for 2 min at 64°C. Total RNA was extracted from the lysate using the hot-phenol method as described previously (**Sharma et al., 2010**). For northern blot analysis, 5–10 µg of total RNA in Gel Loading Buffer II (GLII, Ambion) was loaded per lane on 6% polyacrylamide (PAA)/7 M urea denaturing gels in 1× TBE buffer. Following electrophoretic separation, RNA was transferred to Hybond-XL membranes (GE Healthcare) by electroblotting. Transferred RNA was then cross-linked to the membrane with ultraviolet light to the membrane and hybridized with γ$^{32}$P-ATP end-labeled DNA oligonucleotides (**Supplementary file 1c**) in Roti Hybri-quick (Roth) at 42°C overnight. Membranes were then washed 20 min each at 42° C in 5×, 1×, and 0.5× SSC (saline-sodium citrate) + 0.1% SDS, dried, and exposed to a PhosphorImager screen. Screens were scanned using a FLA-3000 Series PhosphorImager (Fuji) and bands were quantified using AIDA software (Raytest, Germany).

## Total protein sample analysis by SDS-PAGE and western blotting

Analysis of protein expression in *C. jejuni* was performed by SDS-PAGE and western blotting. Bacterial cells were collected from cultures in mid-log phase (OD$_{600}$ 0.4–0.5) by centrifugation at 11,000 *g* for 3 min. Cell pellets were resuspended in 100 µl of 1× protein loading buffer (62.5 mM Tris-HCl, pH 6.8, 100 mM DTT, 10% (v/v) glycerol, 2% (w/v) SDS, 0.01% (w/v) bromophenol blue) and boiled for 8 min. For analysis of total proteins, 0.05–0.1 OD$_{600}$ of cells were loaded per lane on a 12% SDS-polyacrylamide gels. Gels were stained with PageBlue (Thermo Fisher Scientific, #24620). For western blot analysis, samples corresponding to an OD$_{600}$ of 0.05–0.1 were separated on 12% SDS-PAA gels and transferred to a nitrocellulose membrane by semidry blotting. Membranes were blocked for 1 hr with 10% (w/v) milk powder in TBS-T (Tris-buffered saline-Tween-20) and then incubated overnight with primary antibody (monoclonal anti-FLAG, 1:1000; Sigma-Aldrich, #F1804-1MG; or anti-GFP, 1:1000, Roche #11814460001 in 3% bovine serum albumin [BSA]/TBS-T at 4°C. Membranes were then washed with TBS-T, followed by 1 hr incubation with secondary antibody (anti-mouse IgG, 1:10,000 in 3% BSA/TBS-T; GE Healthcare, #RPN4201). All antibodies were dissolved in 3% BSA/TBS-T. After washing, the blot was developed using enhanced chemiluminescence reagent and imaged using an ImageQuant LAS-4000 imager (GE). Bands were quantified using AIDA software. As a loading control, a monoclonal antibody specific for GroEL (1:10,000; Sigma-Aldrich, #G6532-5ML) with an anti-rabbit IgG (1:10,000; GE Healthcare, #RPN4301) secondary antibody was used to probe membranes after FLAG/GFP.

## Rifampicin RNA stability assays

To determine the stability of CJnc180 and CJnc190 in *C. jejuni* NCTC11168 WT and Δ*rnc*, strains were grown to an OD$_{600}$ of 0.5 (mid-log phase) and treated with rifampicin to a final concentration of 500 µg/ml. Samples were harvested for RNA isolation as described above at indicated time points following rifampicin addition (2, 4, 8, 16, 32, 64 min). After isolation, residual DNA was removed from RNA by treatment with DNase I (Thermo Fisher Scientific) according to the manufacturer's recommendations. Ten micrograms of each DNase I-degested RNA sample was used for northern blot analysis as detailed above.

## Primer extension analysis of RNA 5′ ends

Total RNA was extracted from bacteria in log phase as described above. RNA was digested with DNase I (Thermo Fisher Scientific) to remove DNA, and then 5–10 µg of RNA was added to a total volume of 5.5 µl with H$_2$O, denatured, and snap-cooled on ice. A 5′-end $^{32}$P-labeled DNA oligonucleotide complementary to the RNA of interest was then added (*Supplementary file 1c*) and annealed by heating to 80°C, followed by slow cooling (1°C per min) to 42°C. A master mix with reverse transcriptase (RT) buffer and 20 U Maxima RT (Thermo Fisher Scientific) was added and the reaction was allowed to proceed for 1 hr at 50°C. Reactions were stopped with 12 µl GLII (Ambion, 95% (v/v) formamide, 18 mM EDTA, and 0.025% (w/v) SDS, xylene cyanol, and bromophenol blue). A sequencing ladder was also constructed using the DNA Cycle sequencing kit (Jena Bioscience) according to the manufacturer's instructions with the CJnc180/190 region amplified with primers CSO-0354/0355 from genomic DNA (NCTC11168 wild type) as template and the same radioactively labeled primer was used for the reverse transcription reaction. Reactions were separated on 6% or 10% PAA-urea sequencing gels, which were then dried and exposed to a PhosphorImager screen, and then scanned (FLA-3000 Series, Fuji). The following primers were used for primer extension: CJnc190 – CSO-0185; CJnc180 – CSO-0188.

## RACE analysis of RNA 3' ends

Total RNA from *C. jejuni* WT grown to mid-log phase was used for RACE analysis of the 3′ ends of each sRNA. Briefly, 2 µl of 10× Antarctic Phosphatase buffer (NEB), 1 U Antarctic Phosphatase, and 10 U SUPERase•In RNase inhibitor was added to 7.5 µg of denatured/snap-cooled RNA in a 20 µl reaction and incubated for 1 hr at 37°C. The reaction was then made up to 100 µl and extracted with an equal volume of phenol:chloroform:isoamyl alcohol (PCI) in a Phase-Lock gel tube (5PRIME). RNA was then precipitated with 7.5 µg GlycoBlue (Ambion) and 2.5 vol. 30:1 Mix. The RNA was dissolved in water, 250 pmol RNA adaptor E1 (CSO-4916) was added, and the mixture was denatured and snap-cooled. To this was added 2 µl DMSO, 2 µl 10× T4 RNA ligase buffer (NEB), 20 U T4 RNA ligase

enzyme, and 10 U SUPERase•In. Following ligation overnight at 16°C, the RNA was extracted with PCI and precipitated as described above. The RNA was then dissolved in water, denatured and snap-cooled, and subjected to reverse transcription with Maxima reverse transcriptase and DNA oligo E4 for 5 min at 50°C, 1 hr at 55°C, and 15 min at 70°C. RNA was then removed by digestion with 5 U RNase H for 22 min at 37°C. Two microliters of this reaction was then used as template for PCR using *Taq* DNA polymerase (NEB), DNA oligo E4 (CSO-4720), and either CSO-1380 (CJnc190) or CSO-1973 (CJnc180). Cycling conditions were as follows: 5 min at 95°C, 35 cycles of 95°C for 30 s – 57°C for 30 s – 72°C for 45 s, and 10 min at 72°C. Amplification was checked on a 2% agarose/1× TAE gel, reactions were cleaned up with the NucleoSpin Gel and PCR Clean-up Kit (Macherey-Nagel), and ligated to pGEM-T Easy (Promega) according to the manufacturer's instructions. For CJnc180, the inserts of nine white clones were sequenced, and for CJnc190, the inserts of ten white clones were sequenced with primers REV or UNI61.

## In vitro transcription and 5′ end labeling of RNAs

PCR with Phusion DNA polymerase was used to generate DNA templates containing the T7 promoter sequence using oligonucleotides and DNA templates listed in *Supplementary file 1g*. Transcription of RNAs in vitro by T7 RNA polymerase was then carried out using the MEGAscript T7 kit (Ambion) according to the manufacturer's instructions. RNAs were then checked for quality by electrophoresis on a PAA-urea gel, dephosphorylated with antarctic phosphatase (NEB), 5′ end-labeled ($\gamma^{32}$P) with polynucleotide kinase (PNK, Thermo Fisher Scientific), and purified by gel extraction as previously described (*Papenfort et al., 2006*). Sequences of the resulting T7 transcripts are listed in *Supplementary file 1g*.

## Electrophoretic mobility shift assays

Gel shift assays were performed as described previously (*Pernitzsch et al., 2014*). Briefly, 5′ end-radiolabeled RNA (0.04 pmol) was denatured (1 min, 95°C) and cooled for 5 min on ice. Yeast tRNA (1 µg, Ambion) and 1 µl of 10× RNA structure buffer (final concentration 10 mM Tris, pH 7, 100 mM KCl, 10 mM MgCl$_2$) was then mixed with the labeled RNA. Unlabeled RNA (2 µl diluted in 1× structure buffer) was added to the desired final concentrations (0 nM, 10 nM, 20 nM, 50 nM, 100 nM, 200 nM, 500 nM, or 1 µM). Binding reactions were incubated at 37°C for 15 min. Before loading on a pre-cooled native 6% PAA, 0.5× TBE gel, samples were mixed with 3 µl native loading buffer (50% (v/v) glycerol, 0.5× TBE, 0.2% (w/v) bromophenol blue). Gels were run in 0.5× TBE buffer at 300 V and 4°C. Gels were dried, exposed to a PhosphorImager screen, and then scanned (FLA-3000 Series, Fuji).

## Inline probing

Inline probing assays for RNA structure and binding interactions in vitro were performed essentially as described previously (*Pernitzsch et al., 2014*). Five-prime end-labeled RNAs (0.2 pmol, see above) in 5 µl water were mixed with an equal volume of 2× Inline buffer: 100 mM Tris-HCl, pH 8.3, 40 mM MgCl$_2$, and 200 mM KCl and incubated for 40 hr at room temperature to allow spontaneous cleavage. Reactions were stopped with an equal volume of 2× colorless loading buffer (10 M urea and 1.5 mM EDTA, pH 8.0). Reactions were separated on 6% or 10% PAA-urea sequencing gels, which were dried and exposed to a PhosphorImager screen. RNA ladders were prepared using alkaline hydrolysis buffer (OH ladder) or sequencing buffer (T1 ladder) according to the manufacturer's instructions (Ambion).

## RNase III cleavage assays

In vitro-transcribed pre-CJnc180 was 5′ end-labeled as described for Inline probing and electrophoretic mobility shift assay (EMSA) and subjected to RNase III cleavage assays as follows. Labeled pre-CJnc180 (0.2 pmol) was briefly denatured and snap-cooled on ice, followed by the addition of structure buffer to a final concentration of 1× and yeast tRNA to a final concentration of 0.1 mg/ml. Where necessary, unlabeled mature CJnc190 (0.2 or 2 pmol) was denatured and snap-cooled separately and added to reactions. Reactions were pre-incubated at 37°C for 10 min, followed by the addition of RNase III (NEB; 1/625 U) and further incubation at 37°C for 5 min to allow limited cleavage. Reactions were stopped by the addition of an equal volume of GLII and separated on a 10% PAA-urea sequencing gel, which was then dried and exposed to a PhosphorImager screen. RNA ladders were

prepared using alkaline hydrolysis buffer (OH ladder) or sequencing buffer (T1 ladder) according to the manufacturer's instructions (Ambion).

## In vitro translation

In vitro translation of target mRNA reporter fusions in the presence and absence of sRNAs was carried out using the PURExpress system (NEB). An in vitro-transcribed RNA including the *ptmG* 5′ leader (including RBS and CJnc190-binding site) and first 10 codons fused to *gfpmut3* (*ptmG(10th)-gfp*) was used as template for translation (*Supplementary file 1g*). For each reaction, 4 pmol of denatured template RNA was incubated either alone or with equimolar (1×), 10×, or 50× ratios of sRNA species for 10 min at 37°C. In vitro translation components were then added, and reactions were incubated a further 2 hr at 37°C. Reactions were stopped with an equal volume of 2× protein loading buffer. One half of the reaction was analyzed by western blotting on 12% SDS-PAA gels with an antibody against GFP, and the second half was loaded on a second gel which was stained with PageBlue after electrophoresis, as a loading control.

## dRNA-seq data

Processed primary transcriptome data generated by dRNA-seq for *C. jejuni* NCTC11168 (*Dugar et al., 2013*) was retrieved from the NCBI Gene Expression Omnibus (GEO) using the accession GSE38883, and was inspected using the Integrated Genome Browser (bioviz.org) (*Freese et al., 2016*).

## Mass spectrometry

Potential targets of CJnc180/190 were identified by mass spectrometry (MS/MS) analysis of gel-excised, trypsinized protein bands, performed by the Bioanalytical Mass Spectrometry lab at the Max Planck Institute for Biophysical Chemistry (https://www.mpibpc.mpg.de/urlaub) according to published standard protocols. Proteins were separated by SDS-PAGE (12% PAA) and stained with PageBlue protein staining solution (Thermo Fisher Scientific) before analysis.

## Acknowledgements

This work was supported through a grant within the Bavarian Research Network bayresq.net (to CMS). We are grateful to Uwe Plessmann and Prof. Dr. Henning Urlaub for acquiring and analyzing mass spectrometry data as well as Philipp Kible for technical assistance. We thank Dr. Isabelle Iost and Prof. Dr. Fabien Darfeuille for feedback on RNase III assays as well as Sharma lab members for critical comments on the manuscript.

## Additional information

### Funding

| Funder | Grant reference number | Author |
|---|---|---|
| Bavarian Research Foundation | | Cynthia Mira Sharma |

The funders had no role in study design, data collection and interpretation, or the decision to submit the work for publication.

### Author contributions

Sarah Lauren Svensson, Conceptualization, Data curation, Formal analysis, Investigation, Methodology, Validation, Visualization, Writing – original draft, Writing – review and editing; Cynthia Mira Sharma, Conceptualization, Funding acquisition, Project administration, Supervision, Writing – original draft, Writing – review and editing

### Author ORCIDs

Sarah Lauren Svensson http://orcid.org/0000-0002-3183-6084
Cynthia Mira Sharma http://orcid.org/0000-0002-2321-9705

Decision letter and Author response
Decision letter https://doi.org/10.7554/eLife.69064.sa1
Author response https://doi.org/10.7554/eLife.69064.sa2

## Additional files

### Supplementary files
• Transparent reporting form
• Supplementary file 1. Mass spectrometry data; all strains, oligonucleotides, and plasmids used in this study, and details for mutant construction.

### Data availability
All data generated or analysed during this study are included in the manuscript or are provided as supporting data files.

The following previously published datasets were used:

| Author(s) | Year | Dataset title | Dataset URL | Database and Identifier |
|---|---|---|---|---|
| Dugar G, Herbig A, Förstner KU, Heidrich N, Reinhardt R, Nieselt K, Sharma CM | 2013 | High-Resolution Transcriptome Maps Reveal Strain-Specific Regulatory Features of Multiple Campylobacter jejuni Isolates | https://www.ncbi.nlm.nih.gov/geo/query/acc.cgi?acc=GSE38883 | NCBI Gene Expression Omnibus, GSE38883 |

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
