## [Editor Report]

*Campylobacter jejuni* is a serious food-borne pathogen and understanding how the various products necessary for its pathogenesis are regulated is a key step in preventing its growth and/or treating disease. Here, Svensson and Sharma examine the complex pathway that leads to the maturation of two complementary regulatory RNAs and how one of the RNAs antagonizes the other to relieve repression of a virulence-related gene.

---

## [Decision Letter]

**Decision letter after peer review:**

Thank you for submitting your article "RNase III-mediated processing of a *trans*-acting bacterial sRNA and its *cis*-encoded antagonist" for consideration by *eLife*. Your article has been reviewed by 3 peer reviewers, one of whom is a member of our Board of Reviewing Editors and the evaluation has been overseen by Gisela Storz as the Senior Editor. The reviewers have opted to remain anonymous.

Essential revisions:

1) Strengthen the claims of Figure 6 (on 180 antagonizing 190) given that this is a major claim of the study.

2) Revise the manuscript to improve clarity and presentation (all three of the reviewers comment on this).

*Reviewer #1:*

The manuscript describes the mechanisms of biogenesis of two antisense sRNAs by RNase III in C. jejuni, CJnc180 and CJnc190, as well as the specific post-transcriptional activity of CJnc190 on ptmG. The study provides thorough experimental support of (i) binding of CJnC190 to repress translational of ptmG, (ii) RNAse III processing to produce mature CJnc190 and CJnc180 transcripts, (iii) location and contribution of CJnc180/190 promoters and 3' ends, and (iv) mechanisms of RNase III cleavage of CJnc180 and CJnc190. Notably, this study proposes a novel cis-sRNA processing mechanism of CJnc180 in which base pairing with antisense sRNA CJnc190 facilitates proper cleavage by RNase III. Overall, this well constructed and informative study provides impactful knowledge that furthers the field of regulatory RNAs.

The readability of the manuscript should be improved as outlined in the following details.

General:

– Reconsider main text and figures vs. supplemental information and figures selection to allow reader to focus on most important results without distraction of uninteresting data. Manuscript is currently quite long and could benefit from placing details in supplemental, such as lines 141-152 could be shortened to 1 sentence with details in supplemental. A supplementary information section, along with supplemental figures could be useful for readers who would like additional depth.

– Clear strain description or table in supplemental information would aid reader in interpreting results.

Introduction:

– Careful revision of nomenclature. Suggest providing clear definitions of "sponge RNAs" versus "antisense RNAs" that can explain why CJnc180/190 have been categorized as such.

– Much of the discussion details (lines 435-475) belong in the introduction to provide the reader with clear knowledge of what is already known in the field. Suggest including these details in the introduction to be more transparent of contribution to the field. As it stands, the introduction reads as overstating the significance of the work.

– Need description of what is known about RNase III normal function, in addition comments on partial cleavage may be added as well.

Results:

– Lines 140-152 could be supplemental information.

– In addition, lines 162-165 may be moved to a supplemental information section.

– Figure 1A: provide nucleotide coordinates for mature transcripts and promoters. Could also use a brief description in caption what the addition/removal of TEX is expected to do to the dRNA-Seq results.

– Figure 2E: Missing significance analysis or annotation.

– Figure 5B: Provide the pre-processed expected transcript length, as this gel is a bit confusing to read without those as a reference for each truncated form.

– Since the novelty of this paper is really the RNase III degradation mechanism by RNA-RNA interactions, a cartoon figure of the proposed mechanism would really aid understanding.

Discussion:

– As stated above, manuscript can benefit from moving the discussion of the role of RNase III to the introduction to make it easier to understand.

– Provide additional clarity on the statements regarding the toxin-antitoxin system origins discussed in line 526.

– Paper can benefit from a summary figure that details the aspects of the mechanism that is being outlined.

*Reviewer #2:*

*Campylobacter jejuni* is serious food-borne pathogen and understanding how the various products necessary for pathogenesis are regulated is a key step in preventing its growth and/or treating disease. Here, Sharma and coworkers demonstrate the complex pathway that leads to the maturation of two complementary regulatory RNAs and how one of the RNAs antagonizes the other to relieve repression of a virulence-related gene. The work is detailed and convincing, and provides a reference point for the roles of regulatory RNAs in *C. jejuni* as well as other bacteria. Future work will be needed to better understand when each of these RNAs is best expressed and processed into active form, and to fully support the idea that one RNA acts as an antagonist for the other.

1. As noted above, this is a complex paper with a lot going on. A model at the end or possibly the beginning would certainly help, even if it leaves some questions unanswered (processing beyond the RNAse III cutting, expression conditions). The initial part (Figure 1, 2; defining 190 for repression of ptmG) is reasonably easy going, but understanding the various precursors and processing data is more of a challenge.

2. The data at the end of the paper (Figure 6) on 180 antagonizing 190 could be strengthened. At the moment, the only real evidence is the OE data in 6D. Given that this is an important part of the paper, a bit more analysis would be useful. A few thoughts:

a. In the construct that is expressing 190 from only 1 promoter (so decreased amounts), does the presence or absence of 180 have an effect on PtmG-FLAG levels?

b. Do the P1 and P2 versions of 190 change expression during growth in the same fashion that the combined versions do? How does 180 precursor level change with growth stage when 190 is not available? Is there an explanation for why the OE construct works best at exponential (190 levels seem unchanged)?

c. Is it feasible to ask if processed 180 act to relieve processed 190 repression in an rnc mutant? Could that be better done with the PtmG-GFP reporter?

*Reviewer #3:*

In this manuscript the authors describe the biogenesis and the mechanism of action of a pair of cis-encoded sRNAs: CJnc190 and CJnc180. Both RNAs are being processed by RNase III. 5' and 3' ends mapping together with in vitro and in vivo experiments using purified RNase III and rnc deletion mutant demonstrated that the processing of CJnc190 sRNA depended on the formation of an intramolecular duplex, while CJnc180 sRNA processing required the presence of the antisense CJnc190 sRNA. The mature CJnc190 and CJnc180 sRNA specious are 69 and 88 nt long respectively. They also show that mature CJnc190 sRNA represses translation of ptmG via base-pairing and CJnc180 sRNA antagonizes CJnc190 repression acting as a sponge, scavenging CJnc190 sRNA. In addition, they find that two promoters are responsible for the synthesis of CJnc190 sRNA and both transcripts are subject to RNase III processing.

The study represents an enormous amount of work. The data are solid and generally support the overall conclusions. Having said that the manuscript is overwhelming, loaded with too many details which make the reading difficult and in the absence of a bigger picture many times uninspiring.

I suggest:

1) Adding a schematic model figure (including the hybrid formed, the sites of processing etc.).

2) Transfer some nonessential details from the results to the discussion. For example, the characterization of the two transcripts, description of the 5' and 3' mapping- those and other characterizations can be briefly mentioned in the discussion.

3) Expand on the correlation to physiology for the benefit of the bigger picture.

---

## [Author Response]

Essential revisions:1) Strengthen the claims of Figure 6 (on 180 antagonizing 190) given that this is a major claim of the study.2) Revise the manuscript to improve clarity and presentation (all three of the reviewers comment on this).

We thank the editors and three reviewers for their positive assessment of our study and their constructive feedback. We have addressed these comments and suggestions in our enclosed, revised manuscript. We think the changes have overall improved the accessibility of the manuscript and also strengthened our claims regarding the CJnc180 sRNA antagonist.

Experimental additions to the manuscript have focused on Point 1 of the Essential Revisions (CJnc180 sRNA antagonizing CJnc190), as well as differential regulation of the two CJnc190 promoters. Data for this can be found in new Figures 7B, Figure 7 —figure supplement 1C, and Figure 7 —figure supplement 2B-D, as well as some panels within the point-to-point reply below. These data suggest that the antagonistic effect of CJnc180 is stronger when CJnc190 levels are lower (expressed from only a single promoter), supporting that it is the stoichiometry of the two sRNAs that determines the outcome of CJnc180/190 effects on *ptmG* regulation. Our investigations did not identify any conditions so far that specifically regulate any of the CJnc180/190 promoters, which thus remains an open question. Finally, our new data suggests that antagonism might be (in part) RNase III-independent, but does not rule out several mechanisms of competition, including RNase III-mediated decay and transcriptional interference.

For Point 2 (improve clarity and presentation), we have put extensive thought into improving the accessibility of this dense and complex story. We have made minor aesthetic changes throughout for clarity, moved some less important details to Supplementary Figure legends, added several schematics, and have provided a model at the end as requested by the three reviewers (Figure 8). Figure 5 is also now separated into two figures (new Figure 5 and Figure 6) and have removed previous Supplementary Figure 7 (alignment of CJnc180/190 regions from different *Campylobacter*), which is now part of our more extensive comparative analysis of the sRNAs in *Campylobacter spp.* (in revision, submitted version available upon request). We believe that these changes have greatly improved the accessibility of the manuscript.

Reviewer #1:The manuscript describes the mechanisms of biogenesis of two antisense sRNAs by RNase III in C. jejuni, CJnc180 and CJnc190, as well as the specific post-transcriptional activity of CJnc190 on ptmG. The study provides thorough experimental support of (i) binding of CJnC190 to repress translational of ptmG, (ii) RNAse III processing to produce mature CJnc190 and CJnc180 transcripts, (iii) location and contribution of CJnc180/190 promoters and 3' ends, and (iv) mechanisms of RNase III cleavage of CJnc180 and CJnc190. Notably, this study proposes a novel cis-sRNA processing mechanism of CJnc180 in which base pairing with antisense sRNA CJnc190 facilitates proper cleavage by RNase III. Overall, this well constructed and informative study provides impactful knowledge that furthers the field of regulatory RNAs.

We thank the reviewer for their positive feedback on our manuscript.

The readability of the manuscript should be improved as outlined in the following details.General:– Reconsider main text and figures vs. supplemental information and figures selection to allow reader to focus on most important results without distraction of uninteresting data. Manuscript is currently quite long and could benefit from placing details in supplemental, such as lines 141-152 could be shortened to 1 sentence with details in supplemental. A supplementary information section, along with supplemental figures could be useful for readers who would like additional depth.

We agree that several details (e.g., the experimental determination of the mature transcript boundaries) are not critical to the main message of the manuscript. Thus, as suggested by the reviewer, we have moved several details throughout the Results section to figure legends. These are marked in red and/or strikethrough. For example, much of the text from original Lines 141-152 (now at Lines 166-168) was moved to the legend of Figure 1 —figure supplement 4. We also simply reduced or condensed details in other places (e.g., information from original Lines 252-262 (now at Lines 281-293)). For example, some description/details of mapping the locations of precursors and their secondary structures have been condensed, although all mapping information/data is still in the supplementary materials for those interested (Lines 357-367). The 3′ RACE data for both WT and Δ*rnc* has been combined into a single new figure (Figure 1 —figure supplement 3), with less focus on the details in the Results text (e.g., previous Lines 292-309, now visible at Lines 286, 351). Additional changes along similar lines throughout the manuscript are marked in red/strikethrough. Some details from the Results were also moved to the Discussion (see Lines 404-412 and Lines 523-532).

– Clear strain description or table in supplemental information would aid reader in interpreting results.

In the original submission, all strains were already listed and described in a separate supplementary Excel file (as Supplementary Table 3). Maybe this table was not provided for review. This file is now called Supplementary File 1**.** As required by the journal, we have also now included a “Key Resources Table” in our Revision, which references this supplementary file.

Introduction:– Careful revision of nomenclature. Suggest providing clear definitions of "sponge RNAs" versus "antisense RNAs" that can explain why CJnc180/190 have been categorized as such.

Our data suggest the mechanism of CJnc180 antagonism of CJnc190 might be multifactorial and include transcriptional interference, co-processing, and even RNase III-independent “sponging” without degradation (see experiment below for Reviewer 2). Moreover, we do not yet know if CJnc190 might also regulate CJnc180 and its targets (if they exist). We have therefore opted to categorize CJnc180/190 more broadly as “antisense RNAs” or “antagonists”, rather than as sponges. We also more generally refer to all such RNAs with the general term “antagonist”. Examples of these changes can be seen at Lines 36-37, 99, and 508-513.

We do mention the several different names such RNAs have been referred to in the Introduction (Lines 85-86). This nomenclature could be formalized as more examples of antagonizing sRNAs are identified in bacteria and diverse mechanisms of antagonism are described. We are more than open to hearing the reviewer’s suggestions for how to precisely describe such RNAs.

“An open question is whether the two sRNAs simply sequester each other, or play a role in each others’ turnover. RNAs (including sRNAs, mRNAs, or derivatives of other cellular RNAs) that target sRNAs have commonly been termed “sponge” RNAs or competing endogenous RNAs (ceRNAs) (Denham 2020, PMID: 32475775; Grül and Massé 2019, PMID: 30761752). Since the impact of CJnc180 and CJnc190 on each other might be multifactorial and include sequestration from targets, transcriptional interference, or promotion of decay, we suggest currently terming them RNA “antagonists”, and suggest reserving the term “sponge” for those that purely sequester or compete with targets without promoting decay.”

– Much of the discussion details (lines 435-475) belong in the introduction to provide the reader with clear knowledge of what is already known in the field. Suggest including these details in the introduction to be more transparent of contribution to the field. As it stands, the introduction reads as overstating the significance of the work.

As suggested, additional studies and information on the role of RNase III in bacterial sRNA processing was moved to the Introduction (Lines 62-68).

– Need description of what is known about RNase III normal function, in addition comments on partial cleavage may be added as well

The introduction already describes the well-defined roles of RNase III in rRNA processing, CRISPR biogenesis, and cleavage of sRNA-mRNA duplexes (previous Lines 56-63, now Lines 69-70 and 74-75). We have now added a sentence describing background information about RNase III mechanism and specificity (Lines 65-68). In addition, we have noted in the Discussion that RNase III-mediated processing of either sRNA could involve single-stranded nicking (Lines 531-533).

Results:– Lines 140-152 could be supplemental information.

As mentioned above, we have moved most of these details (track changes, Lines 169-179) to the legend of Figure 1 —figure supplements 2 and 3.

– In addition, lines 162-165 may be moved to a supplemental information section.

This information was reduced (now at Lines 188-194), with some details moved to the legend of Figure 1 —figure supplement 4A.

– Figure 1A: provide nucleotide coordinates for mature transcripts and promoters. Could also use a brief description in caption what the addition/removal of TEX is expected to do to the dRNA-Seq results.

We have added the coordinates for CJnc180 and CJnc190 precursor/mature transcript 5’/3’ ends, as well as TSSs, on the x-axis in Figure 1A. The legend also now includes a statement about TEX treatment.

– Figure 2E: Missing significance analysis or annotation.

We have now added this information to Figure 2E in the panel and the legend.

– Figure 5B: Provide the pre-processed expected transcript length, as this gel is a bit confusing to read without those as a reference for each truncated form.

We have now added a schematic, which also incorporates the expected pre-processed CJnc190 lengths, as part of Figure 5B (*left*). We also simplified the labeling of precursors/ lengths on the blot. We hope these changes make the interpretation of the experiment easier, for which our main conclusion is that RNase III-dependent processing of CJnc190 depends on both 5’ and 3’ ends of precursors.

– Since the novelty of this paper is really the RNase III degradation mechanism by RNA-RNA interactions, a cartoon figure of the proposed mechanism would really aid understanding.

We have assumed that the reviewer is referring to either (1) intermolecular interactions between CJnc180 and CJnc190 that mediate CJnc180 processing, or (2) intramolecular base-pairing in CJnc190 that mediates CJnc180-independent processing. Cartoons have been added to Figure 5B and 6B (in addition to the base-pairing schematic already part of Figure 6A), as well as a final model figure (Figure 8). These schematics facilitate understanding the two RNase III-mediated sRNA processing events, as well as the experiments exploring this.

Discussion:– As stated above, manuscript can benefit from moving the discussion of the role of RNase III to the introduction to make it easier to understand.

As suggested by the reviewer, we have moved information from original Lines 435-475 in the Discussion to the Introduction (current Line 73).

– Provide additional clarity on the statements regarding the toxin-antitoxin system origins discussed in line 526.

This was a Reference Manager issue, and was meant to refer only to the *H. pylori* sRNA NikS paper (Eisenbart et al., 2020, PMID: 33002424). We have now removed much of our discussion/speculation about the origins of these sRNAs, as it is now part of a separate manuscript under revision (available upon request) (see Lines 584-591).

– Paper can benefit from a summary figure that details the aspects of the mechanism that is being outlined.

As mentioned above, we have now added/modified an overall model at the end of the manuscript (Figure 8, mentioned at Line 463).

Reviewer #2:Campylobacter jejuni is serious food-borne pathogen and understanding how the various products necessary for pathogenesis are regulated is a key step in preventing its growth and/or treating disease. Here, Sharma and coworkers demonstrate the complex pathway that leads to the maturation of two complementary regulatory RNAs and how one of the RNAs antagonizes the other to relieve repression of a virulence-related gene. The work is detailed and convincing, and provides a reference point for the roles of regulatory RNAs in C. jejuni as well as other bacteria. Future work will be needed to better understand when each of these RNAs is best expressed and processed into active form, and to fully support the idea that one RNA acts as an antagonist for the other.

We thank the reviewer for their positive feedback on our work. Additional experiments (Figure 7B) provide additional evidence that CJnc180 is an antagonist of CJnc190 and affects *ptmG*.

1. As noted above, this is a complex paper with a lot going on. A model at the end or possibly the beginning would certainly help, even if it leaves some questions unanswered (processing beyond the RNAse III cutting, expression conditions). The initial part (Figure 1, 2; defining 190 for repression of ptmG) is reasonably easy going, but understanding the various precursors and processing data is more of a challenge.

As noted for Reviewer 1, we have added a summary model as Figure 8. We also added additional schematics to, e.g., Figures 4A/B, 5B, and 6B.

2. The data at the end of the paper (Figure 6) on 180 antagonizing 190 could be strengthened. At the moment, the only real evidence is the OE data in 6D. Given that this is an important part of the paper, a bit more analysis would be useful. A few thoughts:a. In the construct that is expressing 190 from only 1 promoter (so decreased amounts), does the presence or absence of 180 have an effect on PtmG-FLAG levels?

We have now measured the effect of the CJnc180 antagonist on PtmG-3xFLAG levels when CJnc190 is expressed from only a single promoter (P1 or P2) (new Figure 7B and Figure 7 —figure supplement 1C and 1D, Lines 426-432). This experiment showed that while absence of CJnc180 did not have a significant effect on *ptmG* mRNA or protein levels when CJnc190 was expressed from both P1 and P2, there was a significant increase in mRNA levels (P1 or P2) and protein levels (P2) when CJnc180 expression was abolished when CJnc190 was expressed from only a single promoter. This speaks to the importance of the stoichiometry of the two sRNAs on the outcome of regulation.

b. Do the P1 and P2 versions of 190 change expression during growth in the same fashion that the combined versions do?

CJnc190 appears to be highly stable (Figure 3 —figure supplement 1C) and expressed at all growth phases in wild-type (Figure 7C). In contrast, CJnc180 is relatively unstable (data not shown) and (especially the precursor) decreases as cells enter stationary phase. We have now measured levels of mature CJnc190 in our promoter mutant strains to determine if the sRNA is differentially expressed when expressed from promoter P1 or P2 only (new Figure 7 —figure supplement 3A). This did not reveal any striking differences between CJnc190 expression from either promoter, and both contributed to CJnc190 levels at all growth phases analyzed. This analysis was also performed without CJnc180, but also did not reveal any striking differences (new Figure 7 —figure supplement 3B). These new observations are summarized in the text at Lines 450-451.

To complement this analysis, we also determined if there are marked differences in CJnc180/190 promoter activity at different growth phases. We generated transcriptional GFP reporter fusions to all three CJnc180/190 promoters and measured their expression at three growth phases (new Figure 7 —figure supplement 3C, described at Lines 452-454). While the stability of GFP might obscure downregulation of the promoters (e.g., in stationary phase), we did not observe marked differences in promoter activity between early exponential, mid exponential, and stationary phase, or between CJnc190 P1 or P2. This analysis did show that the CJnc180 promoter is much less active than either CJnc190 promoter under the analyzed growth conditions.

How does 180 precursor level change with growth stage when 190 is not available? Is there an explanation for why the OE construct works best at exponential (190 levels seem unchanged)?

We have now also analyzed CJnc180 levels alone at three growth phases (early exponential, exponential, and stationary, new Figure 7 —figure supplement 2B). Levels of the precursor were much higher than in WT in the presence of CJnc190, and only decreased in stationary phase. This does point to co-processing of CJnc180 together with CJnc190 reducing its stability, although the half-life of the sRNA in WT and Δ*rnc* is comparable (Figure 3 —figure supplement 1C). Therefore, transcriptional interference could also come into play.

Based on all of our data together, the relative levels of the two sRNAs appear to be critical in determining the outcome of *ptmG* regulation. The copy number of CJnc190 is much higher than CJnc180 in log phase, and likely even higher in stationary phase. At later stages of growth, the levels of CJnc180 might be at a point where a two-fold increase does not affect CJnc190.

In addition, post-transcriptional regulation is highly dependent on overall cellular conditions (Gottesman 2004, PMID: 15487940).. Since both sRNAs are processed by RNase III, and are also likely under transcriptional control and possibly regulated degradation by RNases, any growth-phase dependent changes in levels, activity, or availability of these proteins could presumably affect *ptmG* regulation. Specific levels of the target *ptmG* mRNA (or other RNA targets/antagonists) under these conditions could also come into play. This is now mentioned in the text at Lines 546-548.

As we do not have a robust inducible promoter system for *C. jejuni* in hand, and conditions/regulators controlling the sRNAs are still unknown, we examined their interplay in an available “natural” setting where their levels change: different growth phases. Because growth phase has broad effects on cellular physiology and gene expression, we have not made strong conclusions based on the experiment described in Figure 7D.

c. Is it feasible to ask if processed 180 act to relieve processed 190 repression in an rnc mutant? Could that be better done with the PtmG-GFP reporter?

CJnc190 is not processed in an *rnc* deletion mutant and does not accumulate to levels allowing *ptmG* repression (Figure 3). Therefore, we performed the most immediately feasible experiment with the potential to provide insight into this question. In a strain expressing only stable “pre-processed” CJnc190 (5’-end truncation, C-190(Proc)) from unrelated *rdxA*, we added CJnc180 to a second heterologous locus (Author response image 1). This was performed in *trans* to specifically assess the post-transcriptional effect of CJnc180 on CJnc190. We then compared native *ptmG* mRNA levels (the number of available cassettes limited use of an epitope tag) in an RNase III+ or *rnc* deletion background for the strains, and -/+ CJnc180 (Author response image 1).

**Author response image 1. sa2fig1:** Antagonism of CJnc190 by CJnc180 in the presence and absence of RNase III. (**A**) CJnc190(Proc) was inserted at the Cj0046 pseudogene locus under control of its native P1 promoter. CJnc180 was introduced into the unrelated rdxA locus under its native promoter. Analysis was performed in a RNase III+ or Δrnc background. (**B**) Total RNA was analyzed by northern blotting for sRNA expression and ptmG mRNA levels. RnpB RNA was detected as a loading control.

We did observe minor de-repression of *ptmG* mRNA levels when CJnc180 was expressed in this context even in a Δ*rnc* background (*left*, lanes 4 vs 6 and 5 vs 7). However, since the “pre-processed” CJnc190 version is more abundant than CJnc190 in WT and over-represses *ptmG* (Figure 1E and Figure 1 —figure supplement 4A), this might obscure antagonism via sponging by CJnc180. Under native conditions, or if an inducible promoter for CJnc180 would be available, a stronger effect might be observed. Deletion of *rnc* in *C. jejuni* also markedly affects growth. We therefore have chosen to include this experiment only here, rather than as an additional supplementary figure. Future experiments will tease apart the contributions of co-processing and “catalytic” sponging to the antagonistic effect of CJnc180 on *ptmG*.

Reviewer #3:In this manuscript the authors describe the biogenesis and the mechanism of action of a pair of cis-encoded sRNAs: CJnc190 and CJnc180. Both RNAs are being processed by RNase III. 5' and 3' ends mapping together with in vitro and in vivo experiments using purified RNase III and rnc deletion mutant demonstrated that the processing of CJnc190 sRNA depended on the formation of an intramolecular duplex, while CJnc180 sRNA processing required the presence of the antisense CJnc190 sRNA. The mature CJnc190 and CJnc180 sRNA specious are 69 and 88 nt long respectively. They also show that mature CJnc190 sRNA represses translation of ptmG via base-pairing and CJnc180 sRNA antagonizes CJnc190 repression acting as a sponge, scavenging CJnc190 sRNA. In addition, they find that two promoters are responsible for the synthesis of CJnc190 sRNA and both transcripts are subject to RNase III processing.The study represents an enormous amount of work. The data are solid and generally support the overall conclusions. Having said that the manuscript is overwhelming, loaded with too many details which make the reading difficult and in the absence of a bigger picture many times uninspiring.

We thank this reviewer for the overall positive feedback. We agree that this is a very complex story. As detailed in the responses to reviewers 1 and 3, we have made several revisions to the text and Figures and have moved details to the Supplementary Information. We hope this facilitates reading of our manuscript.

I suggest:1) Adding a schematic model figure (including the hybrid formed, the sites of processing etc.)

We agree that this is an important addition, as do the other reviewers, and have now added a model as Figure 8. We also simplified Figure 1A as a starting point, and we also have tried to include more small schematics throughout to aid the reader (*e.g*., top of Figures 4A and 4B). The hybrid formed between the two sRNAs is still part of Figure 6A.

2) Transfer some nonessential details from the results to the discussion. For example, the characterization of the two transcripts, description of the 5' and 3' mapping- those and other characterizations can be briefly mentioned in the discussion.

As suggested, we have removed nonessential details from the results and either moved them to the supplementary figure legends as suggested by Reviewer 1 or to the Discussion, or removed/condensed details. These changes are marked in red and/or strike-through and examples are noted for Reviewer 1 (above).

3) Expand on the correlation to physiology for the benefit of the bigger picture.

To understand under which condition the two CJnc190 promoters (or the CJnc180 antagonist) might be important and gain overall insight into the physiological role of the system, we have also screened total RNA samples from a transcription factor deletion mutant library available in our lab (approximately 35 deletion mutant strains, Svensson and Sharma unpublished) for expression changes of the two sRNAs by northern blotting. This did not reveal a strong candidate for direct transcriptional regulation of either sRNA. However, deletion of the gene encoding the RacR two-component response regulator consistently reduced CJnc190 levels by 50% in log phase (Author response image 2). RacR has been proposed to repress genes involved in fumarate utilization under low oxygen conditions when electron acceptors such as nitrate are present (Apel et al., 2012, PMID: 22343300; van der Stel et al., 2015, PMID: 24707969). However, CJnc180 and CJnc190 P2 promoters were only very mildly, if at all, affected by *racR* deletion (Author response image 2). In addition, an obvious RacR binding motif (van der Stel et al., 2015, PMID: 24707969) does not appear to be present upstream of any of the CJnc180/190 promoters. Therefore, how *racR* deletion affects CJnc190, potentially post-transcriptionally and indirect, remains to be identified. We thus choose to not include these data into the current manuscript and only show it here for the reviewers information.

**Author response image 2. sa2fig2:** Deletion of the gene encoding the RacR two-component response regulator results in lower CJnc190 levels. (**A**) Total RNA from sRNA WT or the indicated mutant strains growing in exponential phase was analyzed by northern blotting for sRNA expression. As a loading control, 5S rRNA was also probed. All strains were in a PtmG-3xFLAG background, allowing parallel western blot detection of target levels with an anti-FLAG antibody. GroEL served as a loading control. (**B**) Activity of CJnc180/190 promoters in the presence and absence of RacR. Expression of promoter GFP reporter fusions to CJnc180 P1, CJnc190 P1/P2, CJnc190 P1, or CJnc190 P2 was measured by western blotting in bacteria growing in exponential phase. Levels were determined in RacR+ or Δ*racR* (-) backgrounds. GroEL served as a loading control.

Furthermore, inspection of a recently published RNA-seq based transcriptomics dataset for another *C. jejuni* WT strain (81-176) under several stress conditions (Avican et al., 2021, PMID: 34078900) did not reveal any that markedly affected levels of either sRNA in any of the examined conditions. In an ongoing project in the lab, we are testing promoter fusions (also for other sRNAs) for regulation by many stress- and host-related compounds, which might in the future provide more information about when each sRNA is expressed. How RNase III levels or activity are regulated in *C. jejuni* is not yet known, although the enzyme appears to prefer manganese over magnesium (Haddad et al., 2013, PMID: 24073828).

Thus, clarifying their physiological roles remains for future studies.

References

Apel D, Ellermeier J, Pryjma M, Dirita VJ, Gaynor EC. Characterization of Campylobacter jejuni RacRS reveals roles in the heat shock response, motility, and maintenance of cell length homogeneity. J Bacteriol. 2012 May;194(9):2342-54. doi: 10.1128/JB.06041-11. Epub 2012 Feb 17. PMID: 22343300; PMCID: PMC3347078.

Avican K, Aldahdooh J, Togninalli M, Mahmud AKMF, Tang J, Borgwardt KM, Rhen M, Fällman M. RNA atlas of human bacterial pathogens uncovers stress dynamics linked to infection. Nat Commun. 2021 Jun 2;12(1):3282. doi: 10.1038/s41467-021-23588-w. PMID: 34078900; PMCID: PMC8172932.

Denham EL. The Sponge RNAs of bacteria - How to find them and their role in regulating the post-transcriptional network. Biochim Biophys Acta Gene Regul Mech. 2020 Aug;1863(8):194565. doi: 10.1016/j.bbagrm.2020.194565. Epub 2020 May 28. PMID: 32475775.

Eisenbart SK, Alzheimer M, Pernitzsch SR, Dietrich S, Stahl S, Sharma CM. A Repeat-Associated Small RNA Controls the Major Virulence Factors of Helicobacter pylori. Mol Cell. 2020 Oct 15;80(2):210-226.e7. doi: 10.1016/j.molcel.2020.09.009. Epub 2020 Sep 30. PMID: 33002424.

Gottesman S. The small RNA regulators of Escherichia coli: roles and mechanisms*. Annu Rev Microbiol. 2004;58:303-28. doi: 10.1146/annurev.micro.58.030603.123841. PMID: 15487940.

Grüll MP, Massé E. Mimicry, deception and competition: The life of competing endogenous RNAs. Wiley Interdiscip Rev RNA. 2019 May;10(3):e1525. doi: 10.1002/wrna.1525. Epub 2019 Feb 13. PMID: 30761752.

Haddad N, Saramago M, Matos RG, Prévost H, Arraiano CM. Characterization of the biochemical properties of Campylobacter jejuni RNase III. Biosci Rep. 2013 Nov 25;33(6):e00082. doi: 10.1042/BSR20130090. PMID: 24073828; PMCID: PMC3839596.

van der Stel AX, van Mourik A, Heijmen-van Dijk L, Parker CT, Kelly DJ, van de Lest CH, van Putten JP, Wösten MM. The Campylobacter jejuni RacRS system regulates fumarate utilization in a low oxygen environment. Environ Microbiol. 2015 Apr;17(4):1049-64. doi: 10.1111/1462-2920.12476. Epub 2014 May 7. PMID: 24707969.